

**Reactive species formed upon interaction of water with fine particulate matter**
**from remote forest and polluted urban air**
Haijie Tong[1*], Fobang Liu[1,2], Alexander Filippi[1], Jake Wilson[1], Andrea M. Arangio[1,3], Yun Zhang[4],
Siyao Yue[5,6,7], Steven Lelieveld[1], Fangxia Shen[1,8], Helmi-Marja K. Keskinen[9,10], Jing Li[11],
Haoxuan Chen[11], Ting Zhang[11], Thorsten Hoffmann[4], Pingqing Fu[7], William H. Brune[12], Tuukka
Petäjä[9], Markku Kulmala[9], Maosheng Yao[11], Thomas Berkemeier[1], Manabu Shiraiwa[13], Ulrich
Pöschl[1]
[1] Multiphase Chemistry Department, Max Planck Institute for Chemistry, 55128 Mainz, Germany
[2] School of Chemical and Biomolecular Engineering, Georgia Institute of Technology, Atlanta, Georgia
30332, USA
[3] École polytechnique fédérale de Lausanne, Lausanne 1015, Switzerland
[4] Institute of Inorganic and Analytical Chemistry, Johannes Gutenberg University, 55128 Mainz, Germany
[5] State Key Laboratory of Atmospheric Boundary Layer Physics and Atmospheric Chemistry, Institute of
Atmospheric Physics, Chinese Academy of Sciences, Beijing, 100029, China
[6] College of Earth and Planetary Sciences, University of Chinese Academy of Sciences, Beijing, 100049,
China
[7] Institute of Surface-Earth System Science, Tianjin University, Tianjin, 300072, China
[8] School of Space and Environment, Beihang University, Beijing, 100191, China
[9] Institute for Atmospheric and Earth System Research / Physics, Faculty of Science, University of Helsinki,
P.O. Box 64, FIN-00014, Helsinki, Finland
[10] Hyytiälä Forestry Field Station, Hyytiäläntie 124, FI-35500 Korkeakoski, Finland
[11] College of Environmental Sciences and Engineering, Peking University, Beijing, 100871, China
[12] Department of Meteorology, Pennsylvania State University, University Park, Pennsylvania 16802, United
States
[13] Department of Chemistry, University of California, Irvine, California 92697-2025, USA
*Correspondence to: Haijie Tong (h.tong@mpic.de)





**Abstract**
Interaction of water with fine particulate matter leads to the formation of reactive species (RS) that may
influence the aging, properties, and health effects of atmospheric aerosols. In this study, we explore the RS
yields of fine PM from remote forest (Hyytiälä, Finland) and polluted urban air (Mainz, Germany and
Beijing, China) and relate these yields to different chemical constituents and reaction mechanisms.
Ultrahigh-resolution mass spectrometry was used to characterize organic aerosol composition, electron
paramagnetic resonance (EPR) spectroscopy with a spin-trapping technique was used to determine the
concentrations $^{\bullet}OH$, $O_2^{\bullet-}$, and carbon- or oxygen-centered organic radicals, and a fluorometric assay was
used to quantify $H_2O_2$ concentration. The mass-specific yields of radicals were lower for sampling sites
with higher concentration of ambient $PM_{2.5}$ (particles with a diameter < 2.5 µm), whereas the $H_2O_2$ yields
exhibited no clear trend. The abundances of water-soluble transition metals and aromatics in ambient $PM_{2.5}$
were positively correlated with the relative fraction of $^{\bullet}OH$ to the totally detected radicals, but negatively
correlated with the relative fraction of carbon-centered radicals. Moreover, we found that the relative
fractions of different types of radicals formed by ambient $PM_{2.5}$ were comparable to the surrogate mixtures
comprising transition metals, organic hydroperoxide, $H_2O_2$, and humic or fulvic acids. Therein humic and
fulvic acids exhibited strong radical scavenging effect to substantially decrease the radical yield of mixtures
comprising cumene hydroperoxide and $Fe^{2+}$. The interplay of transition metals (e.g., iron), highly oxidized
compounds (e.g., organic hydroperoxides), and complexing agents (e.g., humic or fulvic acids), leads to
non-linear concentration dependencies of production and yields of different types of RS. Our findings show
that how the composition of $PM_{2.5}$ influences the amount and nature of RS produced upon interaction with
water, which may explain differences in the chemical reactivity and health effects of particulate matter in
clean and polluted air.



## 1 Introduction

Atmospheric fine particulate matter with a particle diameter < 2.5 μm (PM$_{2.5}$) forms reactive species (RS) upon interaction with water and respiratory antioxidants (Bates et al., 2015;Lakey et al., 2016;Park et al., 2018;Li et al., 2018;Tong et al., 2019). The umbrella term RS comprises reactive oxygen species (e.g., $^{\bullet}$OH, O$_2^{\bullet-}$, $^1$O$_2$, H$_2$O$_2$, and ROOH) as well as C- and O-centered organic radicals (Halliwell and Whiteman, 2004;Sies et al., 2017), which influence the chemical aging of atmospheric aerosols and their interaction with the biosphere (Pöschl and Shiraiwa, 2015;Reinmuth-Selzle et al., 2017;Shiraiwa et al., 2017). For example, Fenton-like reactions of hydroperoxides with transition metal ions contribute to the formation of aqueous-phase radicals including $^{\bullet}$OH (Jacob, 2000;Enami et al., 2014;Anglada et al., 2015;Tong et al., 2016a), enhancing the conversion of organic precursors to secondary organic aerosols (SOA) (Donaldson and Valsaraj, 2010;Ervens et al., 2011;Gligorovski et al., 2015;Gilardoni et al., 2016). Moreover, PM$_{2.5}$ may generate excess concentrations of RS in human airways, causing antioxidant depletion, oxidative stress, and respiratory diseases (Nel, 2005;Cui et al., 2015;Lakey et al., 2016;Qu et al., 2017;Lelieveld and Pöschl, 2017;Rao et al., 2018).

The formation pathways and yields of RS from ambient PM and laboratory-generated SOA have been investigated in a wide range of studies (Valavanidis et al., 2005;Ohyama et al., 2007;Chen et al., 2010;Wang et al., 2011a;Wang et al., 2011b;Verma et al., 2014;Badali et al., 2015;Bates et al., 2015;Verma et al., 2015;Arangio et al., 2016;Tong et al., 2016a;Kuang et al., 2017;Tong et al., 2017;Zhou et al., 2018;Tong et al., 2019;Chowdhury et al., 2019;Fang et al., 2019;Liu et al., 2020). The mass, surface area, and chemical composition of PM were discussed as key factors influencing the reactivity of atmospheric aerosols (Møller et al., 2010;Fang et al., 2015;Jin et al., 2019). Among the substance groups associated with RS formation by PM in water are black carbon (Baumgartner et al., 2014), transition metals (Yu et al., 2018), oxidized aromatic compounds including quinones and environmentally persistent free radicals (Xia et al., 2004;Gehling et al., 2014;Charrier et al., 2014;Xiong et al., 2017), humic-like substances (Lin and Yu, 2011;Page et al., 2012;Fang et al., 2019), and peroxide-containing highly oxygenated organic molecules



(HOMs) (Chen et al., 2010;Wang et al., 2011b;Tong et al., 2016a;Tong et al., 2018;Tong et al., 2019;Fang
et al., 2020;Qiu et al., 2020). Moreover, the humic-like substances and other multifunctional compounds
containing carboxyl, carboxylate, phenolic, and quinoid groups may influence the redox activity of PM via
chelating transition metals (Laglera and van den Berg, 2009;Kostić et al., 2011;Catrouillet et al.,
2014;Gonzalez et al., 2017;Wang et al., 2018c;Win et al., 2018;Wei et al., 2019).

To assess the oxidative potential of ambient PM, the following cellular or acellular assays have been

used: dichloro-dihydro-fluorescein diacetate (DCFH-DA), dithiothreitol (DTT), ascorbic acid (AA),
macrophage, electron paramagnetic resonance (EPR), and surrogate lung fluids (SLF) (Landreman et al.,
2008;Charrier and Anastasio, 2012;Kalyanaraman et al., 2012;Charrier et al., 2014;Charrier and Anastasio,
2015;Fang et al., 2016;Tong et al., 2018;Bates et al., 2019;Fang et al., 2019;Molina et al., 2020;Crobeddu
et al., 2020). However, the interplay of different PM constituents often results in non-additive
characteristics of the RS yields or oxidative potential of PM (Charrier et al., 2014;Lakey et al., 2016;Wang
et al., 2018b). Thus, unraveling the adverse health effects of ambient PM requires systematic investigations
of the RS formation and chemical reactivity of PM from different sources and environments (Shiraiwa et
al., 2017).

The concentration of $PM_{2.5}$ and the composition of airborne organic matter vary considerably from clean

forest to polluted urban environments. For example, the $PM_{2.5}$ concentrations at the Hyytiälä forest site are
typically below 10 µg m$^{-3}$, with organic matter accounting for ~70% (Laakso et al., 2003;Maenhaut et al.,
2011), whereas the $PM_{2.5}$ concentrations in Beijing during winter can reach and exceed daily average values
of 150 µg m$^{-3}$, with organic matter accounting for ~40% (Huang et al., 2014). Moreover, anthropogenic
emissions can enhance the formation of biogenic SOA and HOM as well as the levels of particulate
transition metals, humic-like substances, and PM oxidative potential (Goldstein et al., 2009;Hoyle et al.,
2011;Liu et al., 2014;Xu et al., 2015;Ma et al., 2018;Pye et al., 2019;Shrivastava et al., 2019).

In this study, we compared the RS yields of $PM_{2.5}$ in clean and polluted environments. We used three

approaches to explore the RS formation by $PM_{2.5}$ from remote forest of Hyytiälä (Finland), intermediately
polluted city of Mainz (Germany), and heavily polluted megacity of Beijing (China) (Figure 1). To quantify


the abundances of redox-active PM constituents related to RS formation, we collected ambient $PM_{2.5}$ and
measured the chemical composition of organic matter, the abundance of water-soluble transition metals,
and the yield of radicals and $H_2O_2$ in the liquid phase (Figure 1a). To assess the influence of anthropogenic-
biogenic organic matter interactions on the RS formation by ambient $PM_{2.5}$, we analyzed the radical yield
of SOA generated by oxidation of mixed anthropogenic and biogenic precursors in a laboratory chamber
(Figure 1b). To get insights into the radical formation mechanism of ambient $PM_{2.5}$ in water, we
differentiated the influence of transition metals, organic hydroperoxide (ROOH), water-soluble humic acid
(HA) and fulvic acid (FA) on the radical formation by Fenton-like reactions (Figure 1c).
**2 Materials and methods**
**2.1 Chemicals**
The following chemicals were used as received without further purification: β-pinene (99%, Sigma-
Aldrich), naphthalene (99.6%, Alfa Aesar GmbH&Co KG), cumene hydroperoxide (80%, Sigma-Aldrich),
$H_2O_2$ (30%, Sigma-Aldrich), $FeSO_4 \cdot 7H_2O$ (F7002, Sigma-Aldrich), $CuSO_4 \cdot 5H_2O$ (209198, Sigma-
Aldrich), $NiCl_2$ (98%, Sigma-Aldrich), $MnCl_2$ (≥99%, Sigma-Aldrich), $VCl_2$ (85%, Sigma-Aldrich), NaCl
(443824T, VWR International GmbH), $KH_2PO_4$ (≥99%, Alfa Aesar GmbH&Co KG), $Na_2HPO_4$ (≥99.999%,
Fluka), humic acid (53680, Sigma-Aldrich), fulvic acid (AG-CN2-0135-M005, Adipogen), 5-tert-
Butoxycarbonyl-5-methyl-1-pyrroline-N-oxide (BMPO, high purity, Enzo Life Sciences, Inc.), $H_2O_2$ assay
kit (MAK165, Sigma-Aldrich), ultra-pure water (14211-1L-F, Sigma-Aldrich), 47 mm diameter Teflon
filters (JVWP04700, Omnipore membrane filter), and micropipettes (50 μL, Brand GmbH&Co KG). The
used neutral saline (pH=7.4) consists of 10 mM phosphate buffer (2.2 mM $KH_2PO_4$ and 7.8 mM $Na_2HPO_4$)
and 114 mM NaCl, which was used to simulate physiologically relevant condition.
**2.2 Collection and extraction of ambient fine PM**
Ambient fine particles were collected onto Teflon filters for all sites. The Hyytiälä $PM_{2.5}$ was collected
using a three-stage cascade impactor (Dekati® PM10) at the Station for Measuring Forest Ecosystem-
Atmosphere Relations station (SMEAR II station, Finland) during 31 May-19 July 2017 (Hari and Kulmala,



2005). The Mainz fine PM was collected using a micro-orifice uniform deposit impactor (MOUDI, 122-R,
MSP Corporation) (Arangio et al., 2016) on the roof of Max Planck Institute for Chemistry during 22
August-17 November 2017 and 23-31 August 2018. The Beijing winter $PM_{2.5}$ was collected using a 4-
channel $PM_{2.5}$ air sampler (TH-16, Wuhan Tianhong Instruments Co., Ltd.) in the campus of the Peking
University, an urban site of Beijing, during 20 December-13 January 2016 and 6 November-17 January
2018 (Lin et al., 2015). The sampling time for a single filter sample in Hyytiälä, Mainz, and Beijing are 48-
72, 25-54, and 5-24 h, respectively, depending on the local PM concentrations. More information about the
sampling sites and instrumentation is shown in Table S1. After sampling, all filter samples were put in petri
dishes and stored in a -80 ℃ freezer before analysis. To determine the mass of collected PM, each filter
was weighed before and after the collection using a high sensitivity balance (±10 µg, Mettler Toledo
XSE105DU). In Hyytiälä, the $PM_1$ and $PM_{1-2.5}$ were separately sampled, which were combined and
extracted together to represent $PM_{2.5}$ samples. Mainz PM with cut-size range of 0.056-1.8 µm is taken as a
proxy for $PM_{2.5}$. Particle concentrations in aqueous extracts were estimated to be in the range of 200-6000
µg $mL^{-1}$ (Figure S1).
**2.3 Formation, collection, and extraction of laboratory-generated SOA**
To generate SOA from mixed anthropogenic and biogenic precursors, different concentrations of gas-phase
naphthalene and β-pinene were mixed and oxidized in a potential aerosol mass (PAM) chamber, i.e., an
oxidation flow reactor (OFR) (Kang et al., 2007;Tong et al., 2018). Naphthalene and β-pinene were used
as representative SOA precursors in Beijing and Hyytiälä, respectively (Hakola et al., 2012;Huang et al.,
2019). The concentrations of gas-phase $O_3$ and $^\bullet OH$ in the PAM chamber were ~1 ppm and $~5.0 \times 10^{11}$ cm-
$^3$, respectively. SOA was produced by adjusting the relative concentrations of naphthalene to the sum of it
with β-pinene (i.e., [naphthalene]/([ naphthalene] + [β-pinene])) to be ~9%, ~23%, and ~38%, respectively.
The concentrations of naphthalene and β-pinene were 0.2-0.6 ppm and 1.0-2.5 ppm, respectively, which
were determined on the basis of a calibration function measured by gas chromatography mass spectrometry
(Tong et al., 2018). To investigate the influece of ozone/β-pinene ratios on redox property of SOA, we





measured the aqueous phase radical yields of SOA particles formed from oxidation of ~1 ppm and ~2.5
ppm β-pinene with the same concentration of ozone. With a similar purpose, we measured the radial yields
of SOA formed from oxidation of ~0.2 ppm and ~0.6 ppm naphthalene by the same concenteation of gas-
phase OH radical. The mean radical yields of β-pinene and naphthalene SOA formed at different
concentrations of precursors are compared in Sect. 3.4. The number and size distributions of SOA particles
were meaured using a scanning mobility particle sizer (SMPS, GRIMM Aerosol Technik GmbH&Co. KG).
When the SOA concentration is stable, 47 mm diameter Teflon filters (JVWP04700, Omnipore membrane
filter) were used to collect SOA particles, which were extracted into water solutions within 2 minutes after
the sampling. More information about the SOA formation, characterization, collection, and extraction can
be found in previous studies (Tong et al., 2016a;Tong et al., 2017;Tong et al., 2018;Tong et al., 2019).
**2.4 Surrogate mixtures**
Considering that cumene hydroperoxide (CHP), humic acid (HA), and fulvic acid (FA) have been used as
model compounds mimicking the redox-active substances in biogenic and anthropogenic PM (Lin and Yu,
2011;Ma et al., 2018;Tong et al., 2019), we measured the relative fractions (RF) of different radicals formed
by surrogate mixtures of $CHP+Fe^{2+}+Cu^{2+}+HA+H_2O_2$ to simulate the radical formation by fine PM from
Hyytiälä, Mainz, and Beijing. The $H_2O_2$ was treated as a redox-active constituent preexisting in PM samples
before extraction. The following method was used to make HA or FA solutions. First, 0-1000 $\mu$g mL$^{-1}$ HA
or FA water suspensions were made. Then, the suspensions were sonicated for 3 minutes to accelerate the
dissolution of HA or FA. Afterwards, the sonicated suspensions were centrifuged at 6000 rpm (MiniStar,
VWR International bvba) for 2 minutes. Finally, the supernatants were taken out from the centrifuge tubes
with pipettes and stored in glass vials under 4-8 °C condition before analysis. The HA or FA solutions were
prepared freshly day-to-day. To determine the concentrations of dissolved HA or FA, aliquots of the
supernatants were dried with pure $N_2$ flow (1-2 bar) and weighted with a high sensitivity balance (± 0.01
mg, Mettler Toledo XSE105DU). The concentrations of $Fe^{2+}$, $Cu^{2+}$, HA, and $H_2O_2$ in the surrogate mixtures
are 43 $\mu$M, 3 $\mu$M, 4 mg L$^{-1}$, and 7 $\mu$M, which are based on the measurement of ambient PM extracts ($Fe^{2+}$
and $Cu^{2+}$, Section 2.8) or the estimated abundance in ambient PM (CHP, HA, FA, and $H_2O_2$, SI). To explore

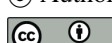

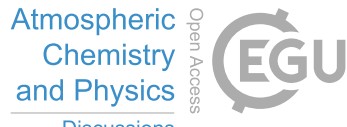

the influence of HA/FA on Fenton-like reactions, the radical formation in the following aqueous mixtures
was also analyzed: CHP+$Fe^{2+}$, CHP+$Cu^{2+}$, CHP+$Cu^{2+}$+HA, CHP+$Cu^{2+}$+FA. The concentrations of $Fe^{2+}$,
$Cu^{2+}$, HA, FA, and $H_2O_2$ in these solutions are 15-300 µM, 15-300 µM, 0-180 µg $mL^{-1}$, 0-180 µg $mL^{-1}$, 0-
300 µM, respectively.
**2.5 Quantification of radicals by EPR**
5-tert-Butoxycarbonyl-5-methyl-1-pyrroline-N-oxide (BMPO) was used as a spin-trapping agent for
detecting different types of radicals formed in the extracts of PM. Ambient PM or laboratory-generated
SOA were extracted from Teflon filters into 10 mM BMPO neutral saline or water solutions by vortex
shaking for ~15 minutes (with Heidolph Reax 1). Around one fourth of each ambient PM filter or a whole
SOA-loaded filter was used for extraction. It was assumed that during the extraction process, most of the
short-lived radicals have reacted with BMPO to form stable adducts.
A continuous-wave electron paramagnetic resonance (CW-EPR) X-band spectrometer (EMXplus-10/12;
Bruker Corporation) was applied for the identification and quantification of radical adducts (Tong et al.,
2016a;Tong et al., 2017;Tong et al., 2018;Tong et al., 2019). In order to increase the signal to noise ratio
of EPR spectra, some of the extracts were concentrated by a factor of 5 - 20 through 15 - 20 minutes drying
under 1-2 bar pure $N_2$ flow. The EPR spectra of BMPO-radical adducts were recorded by setting the
following operating parameters: a microwave frequency of 9.84 GHz, a microwave power of 0.017 mW
(20 dB), a receiver gain of 40 dB, a modulation amplitude of 1 G, a scan number of 50, a sweep width of
100 G, a modulation frequency of 100 kHz, a conversion time of 11 ms, and a time constant of 10 ms.
To average EPR spectra of different $PM_{2.5}$ extracts for each site, the magnetic field values of each
spectrum was transformed to g-values. Then we used the Bruker software, Xenon to do the averaging,
irrespective of the concentrations of $PM_{2.5}$ in extracts. The spin-counting method embedded in Xenon was
applied to quantify radical adducts. The spin-counting method was calibrated using the standard compound
4-hydroxy-2,2,6,6-tetramethylpiperidin-1-oxyl (TEMPOL). To obtain the relative yields of $^\bullet OH$, $O_2^{\bullet-}$, C-
and O-centered organic radicals, EPR spectra were simulated and fitted using the Xenon software before





deconvolution (Arangio et al., 2016;Tong et al., 2018). The weight of assigned species accounts on average
for more than 95% of totally observed radical adducts, which is characterized by the peak area ratios of
corresponding species. EPR spectra with low signal-to-noise ratio introduce uncertainty into the parameters
describing the lineshape of peaks representing radical adducts (Tseitlin et al., 2012), causing a total
quantification uncertainty of 0-19% for the weight and total concentrations of different radical species. The
hyperfine coupling constants used for spectrum fitting are shown in Table S2. More information on the
hyperfine coupling constants of different types of BMPO radical adducts can be found in previous studies
(Zhao et al., 2001;Arangio et al., 2016).
**2.6 Measurement of $H_2O_2$ yields**
We extracted ambient $PM_{2.5}$ from one fourth of each Teflon filter into 1 mL ultra-pure water or neutral
saline by stirring it with a vortex shaker for ~15 minutes. Afterwards, the extracts were centrifuged at 9000
rpm (Eppendorf Minispin) for 3 minutes to remove the insoluble particles. Finally, the concentration of
$H_2O_2$ in the supernatant was measured using the MAK165 assay kit (Yan et al., 2017;Tong et al., 2018). 50
μL of supernatant and 50 μL of a Master Mix solution containing horseradish peroxidase and Amplex Red
substrate were mixed in a 96-well plate. The horseradish peroxidase catalyzed the oxidation of Amplex Red
by $H_2O_2$ to form fluorescent resorufin (Wang et al., 2017), which was consequently quantified using a
microplate reader (Synergy™ NEO, BioTek, excitation at 540 nm and emission at 590 nm) after 30 minutes
of incubation. The concentration of $H_2O_2$ in aqueous PM extracts was determined using an $H_2O_2$ calibration
curve based on standard $H_2O_2$ solutions and also corrected by blank measurements (Tong et al., 2018).
**2.7 Mass spectrometry of organic compounds**
By using a Q-Exactive Orbitrap mass spectrometer (Thermo Fisher Scientific, MA, USA) coupled with an
ultra-high performance liquid chromatography (UHPLC) system (Dionex UltiMate 3000, Thermo
Scientific, Germany) (Wang et al., 2018a;Wang et al., 2019;Tong et al., 2019), we characterized the HOMs
and aromatic compounds in Hyytiälä, Mainz, and Beijing winter fine PM samples in negative ionization
mode. We processed the MS spectrum and UHPLC chromatogram of measured samples through a non-



target screening approach by using the commercially available software SIEVE® (Thermo Fisher Scientific,
MA, USA). Then, we blank-corrected the signals with peak intensity $> 1 \times 10^5$. Afterwards, we used the
following criteria to assign molecular formulae and filter out the irrational ones: (a) the number of atoms
of C, H, O, N, S, and Cl should be in the range of 1-39, 1-72, 0-20, 0-7, 0-4, and 0-2. (b) Atomic ratios of
H/C, O/C, N/C, S/C, and Cl/C should be in the range of 0.3-3, 0-3, 0-1.3, 0-0.8, and 0-0.8, respectively.

The HOMs are defined as formulae fell into the following chemical composition range of $C_xH_yO_z$:

monomers with x = 8–10, y = 12–16, z = 6–12, and z/x > 0.7; dimers with x = 17–20, y = 26–32 and z = 8–
18 (Ehn et al., 2014;Tröstl et al., 2016;Tong et al., 2019). Aromatics in this study are defined to be
compounds with aromaticity index (AI) > 0.5 and aromaticity equivalent ($X_c$) > 2.5, with the parameters
accounting for the fraction of oxygen and sulfur atoms involved in π-bond structures of a compound to be
set as 1 (Koch and Dittmar, 2006;Yassine et al., 2014;Tong et al., 2016b). Beyond this, The relative
abundance of HOMs or aromatic compounds is defined to be the sum chromatographic area of HOMs or
aromatics divided by the sum chromatographic area of all assigned organic compounds, with < 30% of
totally detected organic compounds not assigned (Wang et al., 2018a).
**2.8 Determination of water-soluble transition metal concentrations**
Based on the same extraction method as the $H_2O_2$ analysis in section 2.6, the concentration of five selected
water-soluble transition metal species (Fe, Cu, Mn, Ni and V) in the supernatants of $PM_{2.5}$ extracts was
quantified using an inductively coupled plasma mass spectrometer (ICP-MS, Agilent 7900). These five
transition metal species were chosen for analysis due to their prominent concentrations and higher oxidative
potential (Charrier and Anastasio, 2012). A calibration curve for the ICP-MS analysis was made by
measuring standard multi-element stock solutions (Custom Grade, Inorganic Ventures). An aliquot of the
supernatants was diluted and acidified using a mixture of nitric acid (5%) and hydrofluoric acid (1%), which
was finalized to be 5 mL before analysis. The measured transition metal concentrations were blank-
corrected and shown in corresponding figures. The detection limit of the ICP-MS analysis in this study was
typically < 40 ng $L^{-1}$. The $PM_{2.5}$ samples collected on 2 June, 7 June, 9 June, 12 June in 2017 in Hyytiälä,
on 22 August, 26 August, 28 August, 25 September, 25 October, 14 November in 2017 in Mainz, and all





the 12 PM$_{2.5}$ samples from Beijing winter were used for transition metal analysis. Temporal evolution of
water-soluble transition metal concentrations in water extracts of Mainz PM$_{2.5}$ were also measured, and we
found that the total ion concentration of Fe, Cu, Mn, Ni, and V showed a rapid rise during the first 15 min
(Figure S2a), but at a much slower rate afterwards (Figure S2b).
**3 Results and discussion**
**3.1 Relative yields of different types of radicals from ambient PM$_{2.5}$**
Figure 2a shows the averaged EPR spectra of BMPO-radical adducts in neutral saline extracts of PM$_{2.5}$
samples from Hyytiälä, Mainz (cut-size range 0.056-1.8 µm PM as a proxy), and Beijing. Each spectrum
is composed of multiple peaks attributable to different types of BMPO-radical adducts. The dotted vertical
lines with different colors indicate the peaks attributable to adducts of BMPO with $^\bullet$OH (green), $O_2^{\bullet-}$
(orange), C- (blue) and O-centered organic radicals (purple) (Zhao et al., 2001;Arangio et al., 2016),
respectively. The spectrum of Hyytiälä PM$_{2.5}$ is dominated by peaks attributable to C-centered radicals. In
contrast, the spectrum of Mainz PM$_{2.5}$ comprises strong peaks attributable to $^\bullet$OH and C-centered radicals,
with $^\bullet$OH exhibiting stronger signals. Finally, the spectrum of Beijing winter PM$_{2.5}$ is mainly composed of
four peaks attributable to $^\bullet$OH.

Figure 2b shows the averaged relative fractions (RF) of $^\bullet$OH, $O_2^{\bullet-}$, C- and O-centered organic radicals

generated by multiple PM samples from each site. In line with visual inspection of the spectra in Figure 2a,
the PM$_{2.5}$ from clean forest site generates relatively more C- and O-centered organic radicals but less $^\bullet$OH,
vice versa for the radical yield by PM$_{2.5}$ from polluted areas. Specifically, the mean RF of C- and O-centered
organic radicals, ordered from highest to lowest are: Hyytiälä (66% and 11%) > Mainz (46% and 10%) >
Beijing (39% and 5%). Note that, the significantly higher RF of C-centered radicals than O-centered organic
radicals may be induced by the higher yield and stability of BMPO-C-centered radical adduct in the liquid
phase (De Araujo et al., 2006). Moreover, the C- and O-centered organic radicals may comprise a series of
radicals with different molecular structures, the yields of which are associated with aqueous redox
chemistry of organic matter such as Fenton-like reactions (Arangio et al., 2016;Tong et al., 2018;Tong et



al., 2019). The mean RF of $^{\bullet}OH$, ordered from lowest to highest are: 21% (Hyytiälä) < 38% (Mainz) < 53%
(Beijing). The presence of $^{\bullet}OH$ is related to multiple formation pathways, such as Fenton-like reactions,
thermal or hydrolytic decomposition of peroxide-containing HOMs, and redox chemistry of
environmentally persistent free radicals or aromatic compounds-containing humic-like substances
(Chevallier et al., 2004;Valavanidis et al., 2005;Li et al., 2008;Page et al., 2012;Gehling et al., 2014;Tong
et al., 2016a;Tong et al., 2017;Tong et al., 2018;Tong et al., 2019;Qiu et al., 2020). The mean RF of $O_2^{\bullet-}$
only varies slightly in the range of 2-6%, showing no clear trend and within the range of standard errors in
Figure 2b.
**3.2 Mass-specific and air sample volume-specific yields of RS from ambient $PM_{2.5}$**
Figure 3 shows the mass-specific and air sample volume-specific yields of reactive species (RS) including
radicals, $H_2O_2$, and the sum of radicals and $H_2O_2$ by $PM_{2.5}$ from Hyytiälä, Mainz, and Beijing. The mass-
specific yields of RS are shown in the unit of pmol $\mu g^{-1}$ of $PM_{2.5}$, reflecting the redox activities of $PM_{2.5}$
irrespective of filter loadings. The air sample volume-specific yields of RS are shown in the unit of pmol
$m^{-3}$ of air, indicating that the redox activities of $PM_{2.5}$ scale with atmospheric concentration of $PM_{2.5}$. We
note that, while the more polluted sampling sites led to higher mass loadings, the concentrations of PM in
extracts were found to have a tiny impact on the radical yields (Figure S1c and S1d).
Figure 3a shows that the mass-specific radical yields are negatively correlated with $PM_{2.5}$ mass
concentrations. The mean concentrations of $PM_{2.5}$ are lower to higher in the order of 5 (Hyytiälä) < 16
(Mainz) < 202 $\mu g\ m^{-3}$ (Beijing), whereas the radical yields are in a reverse order of 0.58 > 0.33 > 0.07 pmol
$\mu g^{-1}$. The higher mass-specific radical yield of $PM_{2.5}$ from Hyytiälä may be associated with the higher
abundance of particulate organic matter, which includes quinones and organic hydroperoxides that undergo
thermal, photonic, or hydrolytical dissociation as well as redox chemistry such as Fenton-like reactions to
produce radicals (Badali et al., 2015;Tong et al., 2016a;Tong et al., 2019). More than 70% of $PM_{2.5}$ in
Hyytiälä forest is composed of organic matter (Jimenez et al., 2009;Maenhaut et al., 2011), whereas the
abundances of organic matter in Mainz autumn and Beijing winter $PM_{2.5}$ are ~40% (Jimenez et al.,



2009;Huang et al., 2014), which might in part explain the lower radical yield of these samples. Figure 3a
also shows that the mass-specific $H_2O_2$ yields of $PM_{2.5}$ from Hyytiälä (~2.2 pmol $\mu g^{-1}$), Mainz (~3.4 pmol
$\mu g^{-1}$), and Beijing (~3.4 pmol $\mu g^{-1}$) exhibit a weak positive correlation with $PM_{2.5}$ mass concentrations,
agreeing with previous measurements of the $H_2O_2$ formation by fine PM from different districts of Los
Angeles (Arellanes et al., 2006;Wang et al., 2012) (Figure S4a). The higher $H_2O_2$ yield of urban fine PM
may be associated with its higher abundance of transition metals and aromatic-containing organic matter
(e.g., quinones and humic-like substances), which have been found as redox-active constituents to produce
$H_2O_2$ upon dissolution of ambient PM or laboratory-generated SOA in water (Arellanes et al., 2006;Chung
et al., 2006;Wang et al., 2010;Wang et al., 2012). The weak correlation of mass-specific $H_2O_2$ yields and
$PM_{2.5}$ concentrations reflects the varying redox activity of $PM_{2.5}$ from different regions, which is driven by
the PM source-dependent composition, abundance, and chemistry of redox active substances (e.g.,
transition metals and organic matter).
Figures 3b and S4b show that the air sample volume-specific yields of total RS ($H_2O_2$+radicals) increase
as $PM_{2.5}$ concentrations increase, reflecting a higher RS formation in per cubic meter of polluted urban air.
Specifically, the relative air sample volume-specific yields of $H_2O_2$ (i.e., $[H_2O_2]/([H_2O_2]+[radicals])$),
ordered from lowest to highest are: 78% (Hyytiälä) > 91% (Mainz) > 97% (Beijing), whereas the relative
air sample volume-specific radical yields (i.e., $[radicals]/([H_2O_2]+[radicals])$) are in the reverse order of
22% > 9% > 3%. The relatively stable $H_2O_2$ becomes increasingly important for the reactivity of ambient
$PM_{2.5}$ compared to the more reactive radicals when transitioning from clean to polluted conditions. Due to
its stability, $H_2O_2$ has been found previously to dominate the concentrations of RS formed by $PM_{2.5}$ in liquid
phase with the presence of antioxidants but absence of spin traps (Lakey et al., 2016;Tong et al., 2018).
This study shows a time integral concentration rather than the RS concentration taking into account the
different lifetimes and evolution pathways of radicals and $H_2O_2$. $H_2O_2$ still constitutes the biggest fraction
of RS detected. Of note, the EPR method may not detect all radicals produced but rather a fraction that is
trapped with BMPO before undergoing other radical termination reactions. It is also notable that we
measured the RS yields of PM from three different areas. Further measurements of PM from more locations



may shift the trend of the RS yields in Figure 3 by a certain degree, the extent of which warranty follow-up
studies.

**3.3 Correlation of radical yield with chemical composition of ambient PM$_{2.5}$**

Figure 4 shows how the relative fractions (RF) of C-centered radicals and $^{\bullet}$OH in aqueous extracts of
ambient PM$_{2.5}$ are correlated with the abundance of HOMs, aromatic compounds, and water-soluble
transition metals. Figure 4a shows that the relative abundance of HOMs exhibits a positive correlation with
the RF of C-centered radicals, whereas a negative correlation with the RF of $^{\bullet}$OH. The relative abundance
of HOMs, ordered from lowest to highest are: ~0.2% (Beijing) < ~6% (Mainz) < ~10% (Hyytiälä)  (Tong
et al., 2019), and the RF of C-centered radicals is in the same order of 39% < 46% < 66%, but the RF of
$^{\bullet}$OH are in the reverse order of 53% > 38% > 21%. The higher RF of C-centered radicals formed by PM$_{2.5}$
from less-polluted air is in the same trend as the total mass-specific radical yield of PM$_{2.5}$ from these sites
(Figure 3a), confirming previous results that peroxide-containing HOMs may play an important role in
organic radical formation (Tong et al., 2016a;Tong et al., 2019).
In contrast to HOMs, the relative abundance of aromatic compounds in PM$_{2.5}$ is higher in polluted urban
air compared to clean forest: ~0.2% (Hyytiälä) < ~2% (Mainz) < ~16% (Beijing) (Figure 4b), causing a
positive correlation with the RF of $^{\bullet}$OH, but a negative correlation with the RF of C-centered radicals. The
higher relative abundance of particulate aromatics in Beijing compared to Hyytiälä can be attributed to the
stronger anthropogenic emissions (e.g., from traffic) at the polluted urban site (Jimenez et al., 2009;Zhang
and Tao, 2009;Elser et al., 2016;An et al., 2019). The chemistry of oxygenated aromatic-containing
substances, such as quinones and semiquinones, may enhance the conversion of other RS (e.g., $O_2^{\bullet-}$) into
$^{\bullet}$OH due to redox cycling and interaction with water (Chung et al., 2006;Khachatryan et al., 2011;Fan et
al., 2016).
Similar to the aromatics, the transition metal abundances exhibit a positive correlation with the RF of
$^{\bullet}$OH, but a negative correlation with the RF of C-centered radicals (Figure 4c). The abundance of water-
soluble transition metals in PM$_{2.5}$ from different locations, ordered from lowest to highest are: 13.4



(Hyytiälä) < 19.6 (Mainz) < 27.8 (Beijing) pmol µg⁻¹, and the RF of •OH is in the same order of 21% < 38%
< 53%, whereas the relative fraction of C-centered radicals is in the reverse order of 66% > 46% > 39%.
The consistently higher abundance of water-soluble transition metals and RF of •OH of urban PM$_{2.5}$ may
reflect the importance of Fenton-like reactions in radical formation in polluted air, as H$_2$O$_2$ and
hydroperoxides can be efficiently converted into •OH. Moreover, several studies have reported that metal-
organic interactions may alter the oxidative potential and RS yield of PM under atmospheric and
physiological conditions (Zuo and Hoigne, 1992;Singh and Gupta, 2016;Cheng et al., 2017;Wang et al.,
2018b;Wei et al., 2019;Lin and Yu, 2020). Thus, investigations on the radical chemistry of transition metals
strongly benefit from determination of organic aerosols to illuminate the mechanism of RS formation.
Finally, additional measurements of PM$_{2.5}$ from more locations may shift the correlation of radical yields
and abundances of transition metals and organic matter by a certain degree, the extent of which also
warranty follow-up studies.
**3.4 Radical yield of laboratory-generated SOA**
To investigate the influence of biogenic-anthropogenic organic matter interaction on the formation of
aqueous radicals, we measured the radical yield of SOA generated from oxidation of mixed naphthalene
and β-pinene precursors. Figure 5a shows that the mass-specific radical yields of SOA decrease with
increasing relative concentrations of naphthalene (i.e., [naphthalene]/([naphthalene]+[β-pinene])). As the
relative concentration of naphthalene is increased from 0 to 9, 23, and 38%, the radical yields of SOA
decrease in the order of ~8.4 > ~3.0 > ~2.3 > ~1.9 pmol µg⁻¹. This is because the naphthalene SOA has a
lower radical yield than β-pinene SOA with the same mass concentration in water extracts (Tong et al.,
2016a;Tong et al., 2017;Tong et al., 2018;Tong et al., 2019). Moreover, the mass-specific radical yield of
β-pinene SOA in Figure 5a is the mean value of SOA from ~1 ppm and ~2.5 ppm of β-pinene (see Sect.
2.3). Therein the SOA from ~2.5 ppm β-pinene exhibits higher radical yield (11.5 pmol µg⁻¹) than the SOA
generated from ~1 ppm β-pinene (4.5 pmol µg⁻¹), which may be associated with the increasing partition of
oligomers into the particle phase with higher starting concentration of β-pinene (Kourtchev et al., 2016).



Some oligomers contain peroxide functional groups accounting for a major fraction of HOMs (Krapf et al.,
2016). The radical yield of naphthalene SOA in Figure 5 is the average yields of SOA formed by the
oxidation of ~0.2 ppm and ~0.6 ppm naphthalene (see section 2.3), respectively. Therein the radical yield
of SOA from ~0.2 ppm naphthalene (1.1 pmol $\mu g^{-1}$) is slightly higher than the SOA from ~0.6 ppm
naphthalene (0.8 pmol $\mu g^{-1}$), agreeing with the finding of enhanced oxidative potential of naphthalene SOA
formed under higher oxidant/naphthalene ratio condition (Wang et al., 2018b).

Figure 5b shows that β-pinene SOA mainly generates $^\bullet$OH (~86%), whereas the mixed precursor SOA

and naphthalene SOA mainly generate $O_2^{\bullet-}$ (60-77%) and C-centered radicals (18-34%), which is in line
with our previous findings (Tong et al., 2016a;Tong et al., 2018;Tong et al., 2019). The much lower RF of
$^\bullet$OH formed by mixed precursor SOA (< 10%) may mainly be due to its lower abundance of peroxide-
containing HOMs. It is notable that $PM_{2.5}$ from polluted Beijing contains substantial amount of aromatics
(Figure 4b), but mainly generates $^\bullet$OH upon interaction with water, which seems to contradict our finding
that naphthalene SOA generates $^\bullet$OH only to a small extent. This may be related to the more complex
composition of the ambient PM compared to laboratory-generated SOA. For example, conversion of $O_2^{\bullet-}$
to $^\bullet$OH, $H_2O_2$, and $O_2$ by transition metals or other redox-active PM constituents through Haber-Weiss
reactions or other related redox chemistry (Kehrer, 2000;Tong et al., 2016a) is expected to occur in ambient
samples, but would not be observed in laboratory-generated SOA that does not contain significant fractions
of transition metals.
**3.5 Radical yield of surrogate mixtures comprising transition metals, CHP, HA, FA and $H_2O_2$**
Figure 6a shows the concentration of radicals formed in aqueous mixtures comprising 0-25 $\mu$M cumene
hydroperoxide (CHP), 43 $\mu$M $Fe^{2+}$, 3 $\mu$M $Cu^{2+}$, 4 $\mu g$ $mL^{-1}$ humic acid (HA) and 7 $\mu$M $H_2O_2$, with mixtures
containing 0, 5, and 25 $\mu$M CHP to be treated as surrogates of redox-active constituents in PM from Beijing,
Mainz, and Hyytiälä. As the concentration of CHP is increased from 0 to 25 $\mu$M, the total concentration of
detectable radicals increases from 0.4 to 2.8 $\mu$M, with the relative fractions (RF) of C-centered radicals
increase from 1% to 30%, whereas the RF of $^\bullet$OH and O-centered organic radicals decreases from 72% to

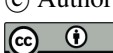



60% and from ~23% to ~8% (Figure 6b), respectively. The higher RF of C-centered radicals but lower RF
of $^\bullet$OH formed at higher concentration of CHP resembles the radical yield of ambient fine PM from cleaner
areas (Figure 2b), which contains a large fraction of HOMs (Tong et al., 2019). Moreover, Figure S5 shows
that adding 75, 100, 150, 200, and 300 μM $H_2O_2$ significantly and linearly ($R^2$=0.95) elevates the $^\bullet$OH
concentration in aqueous mixtures comprising CHP, $Fe^{2+}$, HA, and $H_2O_2$. Thus, the higher RF of $^\bullet$OH in
surrogate mixtures (Figure 6b) compared with ambient PM extracts (Figure 2b) may be due to the choice
of a slightly higher concentration of $H_2O_2$ in the surrogate mixture (7 μM, see SI).
To compare the Fenton-like reactions initiated by different transition metal ions related to ambient PM$_{2.5}$,
we measured the absolute and relative racial yields of aqueous mixtures containing CHP and different
transition metal species, such as $Fe^{2+}$, $Cu^{2+}$, $Mn^{2+}$, or $Ni^{2+}$. We found that $Fe^{2+}$ is most efficient in initiating
Fenton-like reactions (Deguillaume et al., 2005) and the BMPO-radical adduct concentrations varied along
the reaction time (Figure S6). Of note, the abundance, chemical composition, and physicochemical
properties of the redox-active constituents in ambient PM (e.g., transition metals and organic matter) can
be different from the surrogate mixtures, causing partially different radical yields between surrogate
mixtures and ambient PM$_{2.5}$ (e.g., less comparable RF of $^\bullet$OH than the RF of C-centered radicals), which
warrants follow-up studies. To simplify the discussion, we only show the radical yields as mean values
within ~25 minutes of extraction and measurement.
To assess the influence of humic acid on Fenton-like reactions, we measured the radical yields of
mixtures comprising 100 μM CHP, 300 μM $Fe^{2+}$, and 0-180 μg mL$^{-1}$ HA. As the concentration of HA is
increased from 0 to 36 μg mL$^{-1}$, the concentration of total formed radicals decreased by ~52% from 15.5 to
7.4 μM (Figure 6c). This may be associated with the following properties of HA. First, HA exhibits
pronounced iron binding capacity of 32 nmol Fe per milligram of HA, preferentially toward $Fe^{3+}$ rather than
$Fe^{2+}$ (Laglera and van den Berg, 2009;Scheinhardt et al., 2013). Thus, HA may interfere in the redox cycling
of $Fe^{2+}$ and $Fe^{3+}$ by chelating them. The lower concentration of free iron ions may prevent the formation
of radicals via Fenton-like reactions. Second, humic substances have been found to exhibit antioxidant





properties (Aeschbacher et al., 2012), thus the HA used for Figure 6c may act as an RS scavenger, therefore
terminating radical processes and reducing the overall radical concentration. As the HA concentration is
increased further from 36 to 180 µg mL$^{-1}$, the radical concentration is reduced slightly, by less than 20%.
This plateau of radical concentration is accompanied by an increasing RF of C-centered radicals (Figure
6d), indicating that HA may also be involved in more complex radical chemistry with $O_2^{\bullet-}$, $^{\bullet}OH$, or oxygen-
centered organic radicals enhancing carbon-centered radical formation (Shi et al., 2020). In fact, the RF of
C-centered radicals steeply increases from ~19% to ~94% as the HA concentration is increased from 0 to
180 µg mL$^{-1}$, whereas the RF of $O_2^{\bullet-}$ and $^{\bullet}OH$ decreases from ~59% and ~21% to ~3%. The higher RF of
C-centered radicals but lower RF of $O_2^{\bullet-}$ and $^{\bullet}OH$ at higher concentration of HA may be induced by the
reactions of HA with $O_2^{\bullet-}$ and $^{\bullet}OH$. The RF of O-centered organic radicals does not exhibit a consistent
trend and varies within the range of 5-20%. Moreover, we found that the reaction between HA and CHP
(in the absence of Fe ions) produces only a negligible amount of radicals (not shown), which indicates that
HA may mainly influence the radical formation upon interaction with iron ions or radicals formed by
Fenton-like reactions, but does not form prominent amount of radicals by reactions with CHP or through
the decomposition of CHP at the applied concentrations.
Fulvic acid (FA) is another kind of typical atmospheric humic-like substances exerting metal chelating
activity (Graber and Rudich, 2006;Tang et al., 2014). Thus, we also measured the radical yields of the
mixtures comprising CHP, transition metals, and FA. As shown in Figure 6e, the concentration of radicals
formed by mixtures comprising 100 µM CHP, 300 µM Fe$^{2+}$, and FA decreases by ~10% as the
concentration of FA is increased from 6 to 36 µg mL$^{-1}$. Therein the $O_2^{\bullet-}$ is the dominant radial species,
accounting for > 59% of totally formed radicals (Figure 6f). The $O_2^{\bullet-}$ may be generated via multiple redox
reaction pathways such as oxidation of Fe$^{2+}$ or decomposition of organic peroxy radicals (Chevallier et al.,
2004). Figure 6f also shows that RF of $^{\bullet}OH$, $O_2^{\bullet-}$, C- and O-centered organic radicals varies slightly, which
is different from the decreasing radical yield by Fenton-like reaction system containing HA (Figure 6c), but
agreeing with the lower capacity of FA (16.7±2.0 nmol mg$^{-1}$) than HA (32.0±2.2 nmol mg$^{-1}$) in binding





Fe(III) (Laglera and van den Berg, 2009). As the concentration of FA is increased further to 12 µg mL⁻¹,
the observed radical concentration in aqueous mixtures of CHP+Fe²⁺+FA decreases significantly to ~9.6
µM, which may mainly be associated with the formation of Fe-FA complexes and the radical scavenging
effect of FA as discussed for HA above (Wang et al., 1996;Scheinhardt et al., 2013;Yang et al., 2017).
During this process, the RF of C-centered radicals increases for 3-fold to be ~28%, indicating that FA may
also be oxidized by different types of oxidants to form C-centered radicals (Gonzalez et al., 2017), similar
to HA in Figure 6c. As the concentration of FA is increased further to 180 µg mL⁻¹, the concentration of
totally formed radicals decreases further to 7.6 µM, the RF of C-centered radicals increases further to ~36%,
whereas the RF of •OH and O-centered organic radicals decreases significantly to 4-5% and below the
detecting limit, respectively (Figure 6f). Moreover, the Figure S7 indicates that the RF of different radicals
formed by mixtures comprising CHP, Cu²⁺ and FA exhibited a different trend from the mixtures of CHP,
Fe²⁺, and FA, indicating that FA might influence the radical formation by Cu²⁺ initiated Fenton-like
reactions in a efficiency different from the Fe²⁺ initiated Fenton-like reactions.
**4 Conclusions and implications**
In this study, we found that PM₂.₅ levels exhibit a negative correlation with the mass-specific radical yields,
but a weak positive correlation with the H₂O₂ yields. We also found that the mass-specific concentration of
transition metals and relative abundance of aromatic compounds are higher in the urban air than the remote
forest, in the order of Hyytiälä < Mainz < Beijing. The relative fractions (RF) of •OH formed by different
source PM₂.₅ in water is in the same order as the relative abundances of transition metals and aromatics,
indicating that urban fine PM favors the formation of OH radicals upon redox chemistry of transition metals,
aromatics, or transition metal-aromatic interactions in water. The relative abundance of highly oxygenated
organic molecules (HOMs) exhibits a reverse trend compared to aromatics and transition metals, but is in
a positive correlation with the RF of C-centered radicals, confirming the strong association of HOMs with
organic radical formation by PM₂.₅ in water (Tong et al., 2019).



We also measured the radical yield of laboratory-generated SOA from mixing the biogenic SOA
precursor β-pinene and the anthropogenic SOA precursor naphthalene. We found that the relative fractions
of naphthalene SOA of the totally formed SOA significantly influence the amount and types of radicals
formed by the mixed precursor SOA in water with $^\bullet$OH radicals dominating pure β-pinene SOA, Carbon-
centered radicals becoming increasingly dominant as the fraction of naphthalene increases. To get insights
into the Fenton-like reactions in aqueous extracts of ambient $PM_{2.5}$, we investigated the radical formation
by surrogate mixtures comprising cumene hydroperoxide, transition metals, water-soluble humic acid (HA)
or fulvic acid (FA), and $H_2O_2$. We found that HA and FA exhibit different radical scavenging and
antioxidant activity in suppressing the radical formation from Fenton-like reactions.
The synthetic application of ambient $PM_{2.5}$ characterization, chamber simulation, and surrogate mixture
measurement in this study provides a novel approach to investigate the RS chemistry of atmospheric
particles. The direct analysis of ambient $PM_{2.5}$ enables us to find and quantify the key component (e.g.,
HOMs, aromatics, or transition metals) of $PM_{2.5}$ that may influence its reactivity. The investigation of
laboratory-generated SOA enables us to assess the influence of anthropogenic-biogenic organic component
interactions on the radical formation by ambient PM. The measurement of surrogate or aqueous mixtures
of model substances (transition metals, CHP, HA, FA, and $H_2O_2$) enables us to clarify the role of individual
redox active compound as well as their interplays in the radical chemistry of PM, including Fenton-like
reactions, transition metal-organic interactions, or subsequent chain reactions. Based on this systematic
analysis, we quantitatively compared the RS formation mechanism of particulate matter from air ranging
from clean to heavily polluted areas. The higher relative amount of detected radicals and $H_2O_2$ formed by
urban $PM_{2.5}$ can be seen as a measure of higher potential oxidative damage caused by air pollutants in the
epithelial lining fluid of the human respiratory tract. These newly achieved insights enable a better
understanding of the influence of biogenic and anthropogenic emissions on atmospheric chemistry, air
quality, and public health in the Anthropocene (Pöschl and Shiraiwa, 2015;Cheng et al., 2016;Shiraiwa et
al., 2017). Finally, the composition and concentration of organic molecules have been found to influence
its role in transition metal-initiated radical chemistry. For instance, carboxylic acids enhance the oxidative



potential of transition metals, whereas the imidazoles suppress it (Lin and Yu, 2020). Moreover, low
concentration of oxalate forms mono-complexes with $Fe^{2+}$, but high concentration of oxalate scavenges OH
radicals (Fang et al., 2020). Thus, the role of different humic-like substances component in Fenton-like
reactions and its impact on aerosol reactivity have not been fully addressed, which warrants follow up
studies.



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



*Data availability*
The dataset for this paper is available upon request from the corresponding author (h.tong@mpic.de).
*Supporting Information*
Supporting material consists of seven figures and five tables.
*Author contributions*
HT and UP designed the esperiment and wrote up the original draft together with FL. CX, SY, and HK
involved in the collection of ambient particles. HT, FL, AF, and YZ particpated in laboratory measurements
and data analysis. All other co-authors participated in results discussion and manuscript editing.
*AUTHOR INFORMATION*
*Corresponding Author*
*Haijie Tong*
Phone (+49) 6131-305-7040
E-mail: h.tong@mpic.de;
*ORCID*:
Haijie Tong: 0000-0001-9887-7836
Maosheng Yao: 0000-0002-1442-8054
Thomas Berkemeier: 0000-0001-6390-6465
Manabu Shiraiwa: 0000-0003-2532-5373
Ulrich Pöschl: 0000-0003-1412-35570000-0001-9887-7836
*Competing interests*
The authors declare no competing financial interest.
*Acknowledgements*





This work was funded by the Max Planck Society, ACTRIS, ECAC, the Finnish Centre of Excellence under
Academy of Finland (projects no. 307331 and 272041). Siegfried Herrmann and Steve Galer from Climate
Geochemistry Department of Max Planck Institute for Chemistry are gratefully acknowledged for ICP-MS
analysis. Technical staffs at SMEARII station are acknowledged for the impactor maintenance. MS
acknowledges funding from the National Science Foundation (CHE-1808125) and the Japan Society for
the Promotion of Science (JSPS; No. 16K12582).


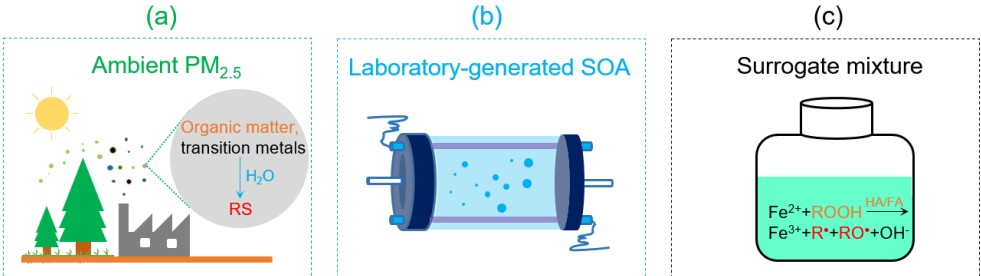


**Figure 1.** Schematic illustration of research approach and comparison of reactive species (RS) formed upon interaction of water with ambient fine particulate matter ($PM_{2.5}$), with laboratory generated secondary organic aerosols (SOA), and in surrogate mixtures. ROOH: organic hydroperoxide. HA: humic acid. FA: fulvic acid. $R^\bullet$ and $RO^\bullet$: C- and O-centered organic radicals, respectively.

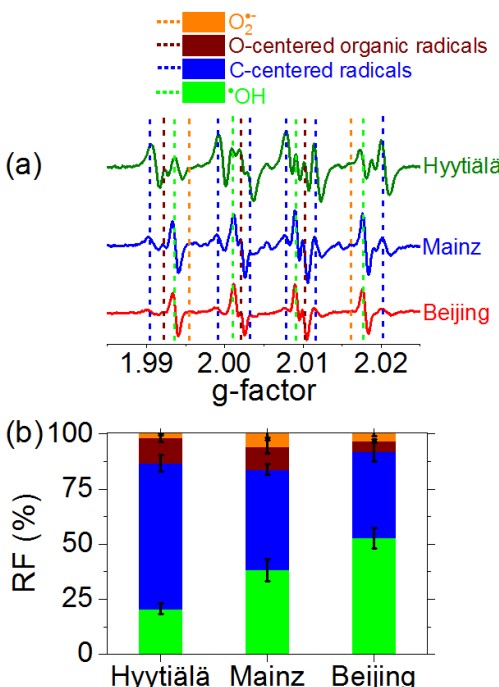

**Figure 2.** (a) EPR spectra and (b) relative fractions (RF) of different types of radicals formed in aqueous extracts of ambient $PM_{2.5}$ from Hyytiälä, Mainz, and Beijing. Dotted vertical lines in (a) indicate peak positions of different radical adducts. The spectra intensity in (a), RF values and error bars in (b) represent arithmetic mean values and standard error (6-13 samples per location).

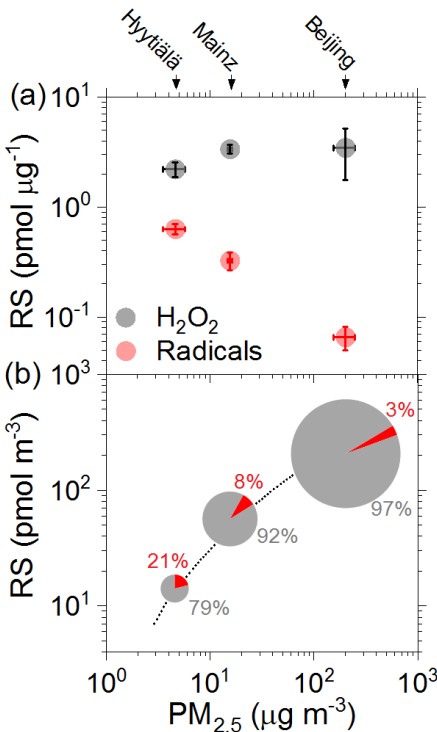

890

**Figure 3.** (a) Mass-specific yield and (b) air sample volume-specific yield of radicals (•) and $H_2O_2$ (◉)
observed upon water interaction of fine $PM_{2.5}$ from Hyytiälä, Mainz, and Beijing plotted against $PM_{2.5}$
concentration. The error bars represent standard errors of the mean (4-12 samples per location). The dotted
line and pie charts are to guide the eye, reflecting the increase of total air sample volume-specific RS yield
(not to scale) and the relative contributions of $H_2O_2$ and radicals.



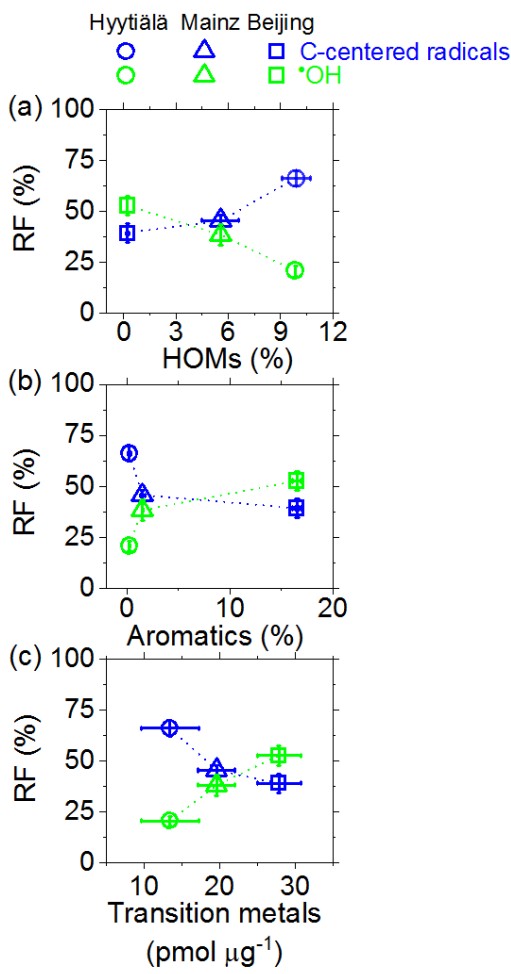

**Figure 4.** Correlation of (a) highly oxygenated organic molecules (HOMs), (b) aromatics, and (c) water-soluble transition metals in ambient $PM_{2.5}$ with relative fractions (RF) of $R^\bullet$ and $^\bullet OH$ observed upon interaction with water. The relative abundances of HOMs and aromatics in (a-b) represent the sum chromatographic area of HOMs or aromatics divided by the sum chromatographic area of all assigned organic compounds. The abundances of HOMs in (a) were adopted from a recent companion study (Tong et al., 2019). The error bars represent standard errors of the mean (4 to 12 samples per location). The dashed lines are to guide the eye.

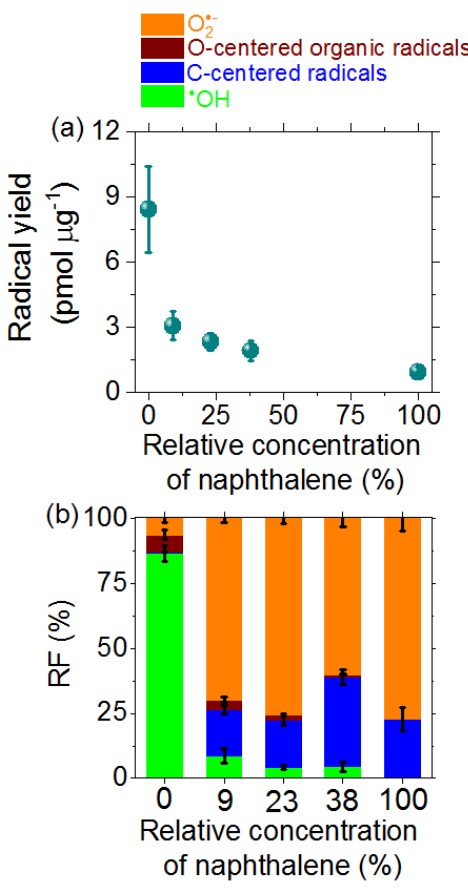

904

**Figure 5.** (a) Mass-specific yields and (b) relative fractions (RF) of radicals formed upon aqueous
extraction of laboratory-generated SOA from different precursors. The relative concentration of
naphthalene represents the relative molar fraction of gas-phase naphthalene to the mixture of naphthalene
and β-pinene. The error bars represent standard errors (4-6 samples per data point).

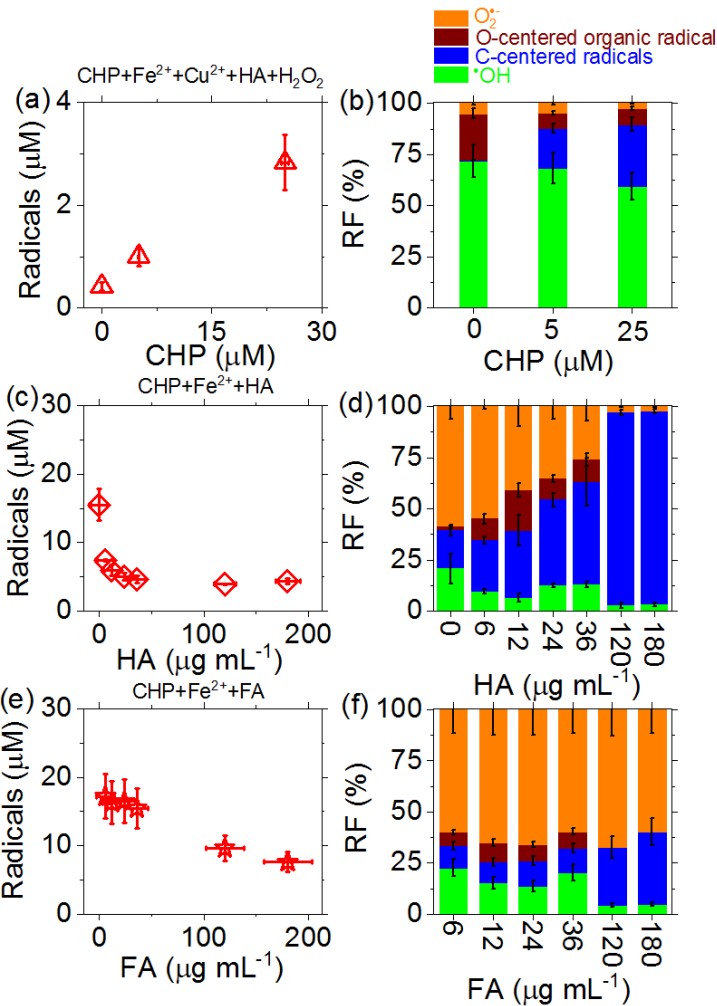

909

**Figure 6.** (a, c, e) Total radical yields and (b, d, f) relative fractions (RF) of different radical types observed in aqueous surrogate mixtures of CHP, $Fe^{2+}$, $Cu^{2+}$, HA, FA, and $H_2O_2$. (a, b): 0-25 μM CHP, 43 μM $Fe^{2+}$, 3 μM $Cu^{2+}$, 4 μg $mL^{-1}$ HA, 7 μM $H_2O_2$ (CHP+$Fe^{2+}$+$Cu^{2+}$+HA+$H_2O_2$). (c, d): 100 μM CHP, 300 μM $Fe^{2+}$, 0-180 μg $mL^{-1}$ HA (CHP+$Fe^{2+}$+HA). (e, f): 100 μM CHP, 300 μM $Fe^{2+}$, 6-180 μg $mL^{-1}$ FA (CHP+$Fe^{2+}$+FA). The error bars represent uncertainties of signal integration of EPR spectra (for y-axis) or experimental uncertainties of the solution concentration (for x-axis). CHP: cumene hydroperoxide. HA: humic acid. FA: fulvic acid.