# Peer review of "Aqueous-phase reactive species formed by fine particulate matter from remote"

_Atmospheric Chemistry and Physics, 2020_

## Referee Comment (RC1) · Anonymous Referee #1 · 10 Nov 2020

**General Comments:** Tong et al. report the field measurement and laboratory experimental investigation on the production of reactive species (RS) including several radicals and hydroperoxides (ROOH) from fine PM dissolved in aqueous solutions. They explored the RS yields of fine PM from remote forest (Hyytiälä, Finland) and polluted urban air (Mainz, Germany and Beijing, China) and related these yields to different chemical constituents and reaction mechanism. They utilized ultrahigh-resolution mass spectrometry for the characterization of organic aerosol composition, electron paramagnetic resonance (EPR) spectroscopy with a spin-trapping technique for the determination of the concentrations $O_2^{\cdot-}$, $HO_2$, OH, carbon- or oxygen-centered organic radicals, and a fluorometric assay for the quantification of $H_2O_2$ concentration. They found that the mass-specific yields of radicals were lower for sampling sites with higher concentration of ambient PM2.5 (particles with a diameter < 2.5 μm), whereas the $H_2O_2$ yields exhibited no clear trend. They also found that the relative fractions of different types of radicals formed by ambient PM2.5 were comparable to the surrogate mixtures comprising transition metals, organic hydroperoxide, $H_2O_2$, and humic or fulvic acids. The experiments were carefully performed, and the analyses were precise. This paper is well-written and the subject showed here are within the scope of ACP. Therefore, I would be happy to accept the manuscript once they address following issues.

**Specific Comments:**

- In the title, abstract and throughout the text, the authors use the words "interaction of water with fine PM". What does this "interaction" really mean? Water itself only contains $H_2O$, $OH^-$ and $H^+$. Although the authors wrote in page 12 "The presence of OH is related to multiple formation pathways, such as Fenton-like reactions, thermal or hydrolytic decomposition of peroxide-containing HOMs, and redox chemistry of environmentally persistent free radicals or aromatic compounds-containing humic-like substances", it is unclear how the observed radicals are actually formed. The key issue is: they already exist and persist in PM? or they are formed just after reactions in water, e.g., $Fe^{2+}$ + ROOH, or an electron-transfer reaction from $OH^-$ to some components in PM? It is fruitful to clarify the meaning of "interaction" and add more discussion on the possible mechanisms of radical formation.

- The authors infer that OH comes from HOMs containing peroxide-groups. However, it has been reported that the $Fe^{2+}$ + ROOH (R=$CH_3$, $C_2H_5$) reaction in water produces RO + $OH^-$, rather than $RO^-$ + OH [Chevallier et al., Atmos. Environ. 2004, 38, 921]. RO would produce $HO_2$. Then, what is a major source of OH-radicals in the present system?

- A PAM chamber experiment would produce atmospherically irrelevant RS species [Peng and

Jimenez, Chem. Soc. Rev., 2020, 49, 2570]. A major RS of PM in ambient air would be a species possessing multiple -OOH formed via autoxidation process involving intramolecular H-abstractions (i.e., HOM). Thus, the high vs. low $O_3$/OH condition critically influence the product identities and the distributions. Does this concentration gap influence the current conclusion? The authors should comment on the issue.

- Can the authors estimate how much $O_2$ is dissolved in solutions? The presence of $O_2$ in solutions could induce autooxidation reactions that give OH-radical signals of EPR [Floyd and Wiseman, Biochim. Biophys. Acta, 1979, 586, 196]. $O_2$ in solutions could also influence the Fenton-like chemistry and HOx cycles in the system [Chevallier et al., Atmos. Environ. 2004, 38, 921]. If the authors perform degassing of solutions (e.g., by Argon) before measurement, are the same results obtained? It would be better to perform such a test to understand the mechanism in more detail.

- C-centered radicals are expected to rapidly react with $O_2$ or recombine in condensed phases. Why were they so abundant (Figures 2 and 4)? What types of C-centered radicals can be assumed? The authors should comment on the issue.

- BMPO-OOH must decompose into other products in solutions (it decreases as a function time in Fig. S6). What are the decomposition products? Do the products influence the RS yields? The authors should comment on the issue.

- Fig. 3 shows $H_2O_2$ is the dominant RS in PM2.5, while radical species contribute just as minor components. What is the main source of the observed high $H_2O_2$? Furthermore, from the viewpoint of adverse health effects of PM inhalation, if $[H_2O_2] \gg$ [radicals] in lung epithelium lining fluid, how important are these radicals? It would be better to add some discussion on these issues.

**Minor Comment:**
- Line 858, "participated"

---

## Referee Comment (RC2) · Anonymous Referee #2 · 18 Nov 2020

Tong et al. report systematic investigations on RS yields of ambient PM in clean and polluted environments using EPR spectroscopy. Furthermore, they found that using the surrogate mixtures can reproduce the relative fractions of different types of radicals formed by ambient PM, which would be useful for future laboratory studies. This ms is generally well written, highlighting the complicated mechanisms responsible for radical formation, but it lacks mechanistic insight into how different chemical compositions in PM generate different relative fractions of RS. I hope to see more detailed discussions about the mechanisms in the text, although it is understood that It may be beyond the scope of this study. Overall, I support this ms to be published in ACP after addressing the following points.

**Major points**

Can you elaborate why naphthalene and beta-pinene were used as representative SOA in Beijing and Hyytiala, respectively? Anthropogenic and biogenic representatives?

For the sample treatment procedure of HA or FA water suspensions, why was the procedure necessary? So, does it mean that a water-soluble fraction of HA or FA was extracted and only analyzed?

Relative fractions of different radicals, particularly organic and OH radicals, seem to be dependent on the origin of PM samples. What would be the most important factor determining such relative fractions?

In Figure 4, why were only transition metals shown in its absolute concentration, whereas the others were in the relative abundance (%)?

Lines 369-373: Why did naphthalene SOA have a lower radical yield than  $\beta$ -pinene SOA with the same mass concentration? This information should be important to generalize the findings of this study.

It seems that consistently the higher relative fraction of C-centered radicals is associated with the lower relative fraction of OH radicals. Does it suggest that OH radicals are consumed by something to generate C-centered radicals? This point deserves detailed discussions in the text.

When HA was replaced by FA, the dominant radical species becomes O2- from C-centered radical. Do you any explanation on the different radical distribution?

Lines 496-498: It is said that "the higher relative amount of detected radicals and H2O2 formed by urban PM2.5 can see as a measure of higher potential oxidative damage", but the totally formed radical amount decreases with PM mass. The statement seems contradicting and needs clarification.

Minor points

Line 236: The -> the

---

## Author Comment (AC1) · 4 Jun 2021

**General Comments**: Tong et al. report the field measurement and laboratory experimental investigation on the production of reactive species (RS) including several radicals and hydroperoxides (ROOH) from fine PM dissolved in aqueous solutions. They explored the RS yields of fine PM from remote forest (Hyytiälä, Finland) and polluted urban air (Mainz, Germany and Beijing, China) and related these yields to different chemical constituents and reaction mechanism. They utilized ultrahigh-resolution mass spectrometry for the characterization of organic aerosol composition, electron paramagnetic resonance (EPR) spectroscopy with a spintrapping technique for the determination of the concentrations $O_2^{\bullet-}$, $HO_2$, OH, carbon- or oxygen-centered organic radicals, and a fluorometric assay for the quantification of $H_2O_2$ concentration. They found that the mass-specific yields of radicals were lower for sampling sites with higher concentration of ambient $PM_{2.5}$ (particles with a diameter < 2.5 μm), whereas the $H_2O_2$ yields exhibited no clear trend. They also found that the relative fractions of different types of radicals formed by ambient $PM_{2.5}$ were comparable to the surrogate mixtures comprising transition metals, organic hydroperoxide, $H_2O_2$, and humic or fulvic acids. The experiments were carefully performed, and the analyses were precise. This paper is well-written and the subject showed here are within the scope of ACP. Therefore, I would be happy to accept the manuscript once they address following issues.

Response: We thank the referee #1 for reviewing and providing positive comments. The point-by-point responses are given below. We have highlighted the changed text in blue.

**Specific Comments:**

In the title, abstract and throughout the text, the authors use the words "interaction of water with fine PM". What does this "interaction" really mean? Water itself only contains $H_2O$, $OH^-$ and $H^+$. Although the authors wrote in page 12 "The presence of OH is related to multiple formation pathways, such as Fenton-like reactions, thermal or hydrolytic decomposition of peroxide-containing HOMs, and redox chemistry of environmentally persistent free radicals or aromatic compounds-containing humic-like substances", it is unclear how the observed radicals are actually formed. The key issue is: they already exist and persist in PM? or they are formed just after reactions in water, e.g., $Fe^{2+}$ + ROOH, or an electron-transfer reaction from OH to some components in PM? It is fruitful to clarify the meaning of "interaction" and add more discussion on the possible mechanisms of radical formation.

Response: Thank you for pointing out the unclear definition of 'interaction', by which we mean the chemical reactions that occur after PM is dissolved in water. For clarity, we replaced the term of '*interaction*' with '*aqueous-phase reactive species formation*' or '*reactive species formation in the aqueous phase*' throughout the manuscript, including the title.

To clarify how the observed radicals are actually formed, we add the following discussion to the main text (lines 294-301):

*Environmentally persistent free radicals (EPFR), are known to pre-exist in $PM_{2.5}$ at mass-specific concentration levels of ~0.2 to ~2 pmol $\mu g^{-1}$, which are an order of magnitude higher than the typical mass-specific aqueous-phase radical yields of ~0.02 to ~0.2 pmol $\mu g^{-1}$ (Arangio et al., 2016;Vejerano et al., 2018;Tong et al., 2019;Chen et al., 2020). While some EPFR may be water-insoluble (Chen et al., 2018), others may directly contribute to the C-centered and O-centered radicals trapped by BMPO or participate in redox reactions yielding $^{\bullet}OH$ and $O_2^{\bullet-}$ radicals (Khachatryan et al., 2011;Arangio et al., 2016). The latter have such short chemical lifetimes that they have to be formed upon dissolution of the investigated samples immediately prior to trapping by BMPO.*

More discussions on the radical formation mechanism are given in the responses below.

The authors infer that OH comes from HOMs containing peroxide-groups. However, it has been reported that the $Fe^{2+}$ + ROOH (R=$CH_3$, $C_2H_5$) reaction in water produces RO + $OH^-$, rather than $RO^-$ + OH [Chevallier et al., Atmos. Environ. 2004, 38, 921]. RO would produce $HO_2$. Then, what is a major source of OH-radicals in the present system?

Response: Thank you for raising this important point. To explain the PM source-dependent formation mechanisms of aqueous-phase radicals, we add the following discussions in the main text (lines 282-293):

*We speculate that hydrolytic or thermal decomposition of ROOH may play a major role in the formation of RS by $PM_{2.5}$ from remote forest locations like Hyytiälä, where large fractions of peroxide-containing HOM have been detected (Mutzel et al., 2015;Tröstl et al., 2016;Tong et al., 2019;Pye et al., 2019;Roldin et al., 2019;Bianchi et al., 2019). ROOH can generate $^{\bullet}OH$ and O-centered organic radicals through decomposition (ROOH $\rightarrow RO^{\bullet} + {}^{\bullet}OH$) and Fenton-like reactions ($Fe^{2+} + ROOH \rightarrow Fe^{3+} + RO^{\bullet} + OH^-$; $Fe^{2+} + ROOH \rightarrow Fe^{3+} + RO^- + OH^{\bullet}$) (Tong*

*et al., 2016). Interconversion of $RO^\bullet$, $R^\bullet$ and $ROO^\bullet$ radicals can lead to the formation of $O_2^{\bullet-}$ and $H_2O_2$ (Chevallier et al., 2004;Tong et al., 2018), which can further react with $Fe^{2+}$ to form $^\bullet OH$ ($Fe^{2+} + H_2O_2 \rightarrow Fe^{3+} + {}^\bullet OH + OH^-$). In $PM_{2.5}$ from urban areas, transition metal ions and HU-LIS are expected to play a major role in aqueous-phase formation and interconversion of $^\bullet OH$, $O_2^{\bullet-}$ and $H_2O_2$ (Lloyd et al., 1997;Valavanidis et al., 2000;Zheng et al., 2013;Hayyan et al., 2016;Lakey et al., 2016;Tan et al., 2016;Kuang et al., 2017;Ma et al., 2018;Li et al., 2019).*

A PAM chamber experiment would produce atmospherically irrelevant RS species [Peng and Jimenez, Chem. Soc. Rev., 2020, 49, 2570]. A major RS of PM in ambient air would be a species possessing multiple -OOH formed via autoxidation process involving intramolecular H-abstractions (i.e., HOM). Thus, the high vs. low $O_3$/OH condition critically influence the product identities and the distributions. Does this concentration gap influence the current conclusion? The authors should comment on the issue.

Response: Thank you for this valuable comment. We agree that there is an $O_3$/$^\bullet OH$ concentration gap from the laboratory chamber experiment to the real ambient air, and the distributions and identities of HOM in SOA depend on the absolute concentrations of $O_3$/OH and the ratio of oxidant to precursor concentration. However, it is not possible for us to resolve the issue in this study and we plan to accommodate the problem in future studies. To clarify this point, we add the following text in the SI (lines 46-50):

*To note, there is a gap between the concentration of gas phase $O_3$ or $^\bullet OH$ in laboratory chamber experiment and ambient air. In the PAM chamber, one might form atmospherically irrelevant RS under the high oxidant conditions (Peng and Jimenez, 2020). The distributions and identities of HOM in SOA depend on the absolute concentrations of $O_3$ or $^\bullet OH$ and also the concentration ratio of oxidant to precursor, which warrants further analysis.*

Can the authors estimate how much $O_2$ is dissolved in solutions? The presence of $O_2$ in solutions could induce autooxidation reactions that give OH-radical signals of EPR [Floyd and Wiseman, Biochim. Biophys. Acta, 1979, 586, 196]. $O_2$ in solutions could also influence the Fenton-like chemistry and $HO_x$ cycles in the system [Chevallier et al., Atmos. Environ. 2004, 38, 921]. If the authors perform degassing of solutions (e.g., by Argon) before measurement, are the same results obtained? It would be better to perform such a test to understand the mechanism in more detail.

Response: Our measurements were conducted under ambient conditions and a degassing was not performed. We assume that the concentration of oxygen in the aqueous phase (~0.29 mM) is determined by Henry's law and constant over reaction time. To identify the influence of aqueous-phase $O_2$ on the radical formation by SOA in water, we also determined the radical yields of β-pinene SOA in degassed water (with ultrapure $N_2$ for ~1 h and keep the $N_2$ exposure during the extraction operation). Indeed, we found that dissolved $O_2$ increases the radical yield of 1 mM SOA by ~20%. In both experiments, the majority of detected radicals were $^\bullet OH$ (RF > 80%). To clarify the potential influence of aqueous-phase $O_2$ on the radical formation by SOA, the following sentences were added to the SI (lines 122-133):

*Aqueous-phase $O_2$ has been suggested to be capable of inducing autooxidation reactions and influencing the Fenton-like chemistry as well as $HO_x$ cycles of organic hydroperoxides in water (Floyd and Wiseman, 1979;Chevallier et al., 2004). We thus compared the radical yields of β-pinene SOA in non-degassed and degassed water (with ultrapure $N_2$ for ~1 h and keep the $N_2$ exposure during the extraction operation). We found that OH radicals are always the major species trapped by BMPO and detected with EPR within both environments. Moreover, ~20% more radicals were observed upon dissolving β-pinene SOA in non-degassed water (1 mM), reflecting the important role of $O_2$ in the radical formation by SOA in water. These experimental results can be explained by our recent modelling analysis, which shows that absence of $O_2$ will lead to the recombination of C-centered radicals $R^\bullet$ and interrupt formation of peroxy radicals and superoxide radicals from $R^\bullet$ (Tong et al., 2017;Tong et al., 2018). In this study, measurements were conducted under ambient, non-degassed conditions. We assume that the concentration of oxygen in the aqueous phase (~0.29 mM) is determined by Henry's law and remain constant over reaction time.*

C-centered radicals are expected to rapidly react with $O_2$ or recombine in condensed phases. Why were they so abundant (Figures 2 and 4)? What types of C-centered radicals can be assumed? The authors should comment on the issue.

Response: To explain the observation of substantial amounts of C-centered radicals, we add the following discussions in the main text (lines 265-271):

*The high yield of C-centered radicals can be explained by rapid trapping of C-centered organic radicals ($R^\bullet$) by BMPO in the liquid phase (De Araujo et al., 2006). In the aqueous extracts, we applied a large excess of BMPO (10 mM of BMPO vs. ~1 µM of trapped radicals), and the estimated pseudo-first-order rate coefficient for $R^\bullet$ reacting with BMPO ($9\times10^5$ $s^{-1}$, (Tong et al., 2018)) is much higher than the estimated $R^\bullet$ recombination rate coefficient ($2.4\times10^3$, (Simic et al., 1969;Tong et al., 2018)). Moreover, rearrangement reactions in water can convert $RO^\bullet$ into $R^\bullet$ (Chevallier et al., 2004), which may warrant further investigation.*

BMPO-OOH must decompose into other products in solutions (it decreases as a function time in Fig. S6). What are the decomposition products? Do the products influence the RS yields? The authors should comment on the issue.

Response: The referee is correct, there are multiple sinks of the BMPO-OOH, including the conversion of BMPO-OOH to BMPO-OH, direct dissociation, and reactions with other reactants (e.g., metal ions and radicals) in the aqueous extracts of PM (Tong et al., 2018). Due to the general limitation of the spin trapping and EPR spectrometry techniques in differentiating different source radicals (e.g., primary versus secondary), we cannot assess the influence of decomposition products of BMPO-radical adducts quantitatively without further studies. To clarify this point, we add the following text in the SI (lines 116-120):

*To note, there are multiple sinks of the BMPO-radical adducts, which include the conversion of BMPO-OOH to BMPO-OH, direct dissociation, and reactions with other reactants (e.g., metal ions and radicals) in the aqueous extracts of PM (Tong et al., 2018). It remains a challenge to assess the influence of decomposition products of BMPO-radical adducts on the radical detection in this study, warranting further studies.*

Fig. 3 shows $H_2O_2$ is the dominant RS in $PM_{2.5}$, while radical species contribute just as minor components. What is the main source of the observed high $H_2O_2$? Furthermore, from the viewpoint of adverse health effects of PM inhalation, if $[H_2O_2] \gg [\text{radicals}]$ in lung epithelium lining fluid, how important are these radicals? It would be better to add some discussion on these issues.

Response: To note, ROS concentrations in lung epithelium lining fluid (ELF) may be different compared to those observed in water because of the presence of enzymes and antioxidants in the

ELF (Tong et al., 2018). The $H_2O_2$ formation by ambient PM has been suggested to be strongly associated with α-hydroxyhydroperoxides, transition metals, and quinones (Arellanes et al., 2006;Charrier et al., 2014;Lakey et al., 2016;Wei et al., 2021). The two-step reduction of $O_2$ with reductive substances (e.g., transition metals and semiquinone, SQ, radicals) to form super-oxide radicals ($SQ/TM^Z + O_2 \rightarrow Q/TM^{Z+1} + O_2^{\bullet}$-) and subsequently $H_2O_2$ ($SQ/TM^Z + O_2^{\bullet}$- $\rightarrow$ $Q/TM^{Z+1} + H_2O_2$) may be the major source of the observed high $H_2O_2$ in this study. Excessive amounts of exogenous $H_2O_2$ not only deplete enzymes and antioxidants, but also act as precursor of $^{\bullet}OH$, inducing further respiratory damages (Halliwell et al., 2000). Indeed, compared to $H_2O_2$, the measured radical concentrations are relatively low, however, the radicals are much more reactive. Moreover, the EPR method may not detect all radicals produced but rather a fraction that is trapped with BMPO before undergoing other radical termination reactions. Therefore, the overall health effect of cumulatively formed number of radicals is unclear and warrants follow-up studies. We add the following text in in lines 322-326 to discuss the source of aqueous-phase $H_2O_2$ in $PM_{2.5}$ extracts:

*The strong increase of $H_2O_2$ with increasing $PM_{2.5}$ concentration is consistent with earlier studies identifying a wide range of redox-active organic and inorganic aerosol components that can produce $H_2O_2$ in the aqueous phase (Gunz and Hoffmann, 1990;Anastasio et al., 1994;Zuo and Deng, 1997;Arellanes et al., 2006;Chung et al., 2006;Hua et al., 2008;Möller, 2009;Wang et al., 2010;Wang et al., 2012;Anglada et al., 2015;Herrmann et al., 2015;Lakey et al., 2016;Tong et al., 2018;Bianco et al., 2020).*

**Minor Comment:**
Line 858, "participated"
Response: Thanks. We corrected the typo.

[revised manuscript text omitted]

---

## Author Comment (AC2) · 4 Jun 2021

**Response to the comments of Anonymous Referee #2**

Tong et al. report systematic investigations on RS yields of ambient PM in clean and polluted environments using EPR spectroscopy. Furthermore, they found that using the surrogate mixtures can reproduce the relative fractions of different types of radicals formed by ambient PM, which would be useful for future laboratory studies. This ms is generally well written, highlighting the complicated mechanisms responsible for radical formation, but it lacks mechanistic insight into how different chemical compositions in PM generate different relative fractions of RS. I hope to see more detailed discussions about the mechanisms in the text, although it is understood that it may be beyond the scope of this study. Overall, I support this ms to be published in ACP after addressing the following points.

Response: We thank the referee #2 for reviewing and providing positive comments. The point-by point responses are given below. We have highlighted changes to the manuscript text in blue.

Major points

Can you elaborate why naphthalene and beta-pinene were used as representative SOA in Beijing and Hyytiälä, respectively? Anthropogenic and biogenic representatives?

Response: Thank you. First, monoterpenes have been found as major volatile organic compounds (VOC) associated with SOA formation in Hyytiälä, and the α- and β-pinene are isomers accounting for > 60% of the total VOC (Kourtchev et al., 2008;Hakola et al., 2012). Moreover, previous studies found that α- and β-pinene exhibit high SOA yield upon oxidation by $O_3$ (Lee et al., 2006;Zhang et al., 2015), and the β-pinene SOA exhibit higher potentials in generating radicals and $H_2O_2$ in water (Tong et al., 2016;Tong et al., 2017;Tong et al., 2018;Tong et al., 2019). Therefore, β-pinene was chosen as representative biogenic precursor for the SOA formation in Hyytiälä. In contrast, naphthalene has been found as key anthropogenic VOC in Beijing. Naphthalene has a high SOA yield from photooxidation and potentially contributes to SOA formation in Beijing (Chan et al., 2009;Huang et al., 2019). Recent studies also indicate that naphthalene SOA contains redox-active substances and may play an important role in the cytotoxicity of Beijing PM (Liu et al., 2020;Han et al., 2020). Thus, we choose naphthalene as representative anthropogenic precursor for the SOA formation in Beijing. We add the following section in the SI (lines 55-67) to elaborate why naphthalene and β-pinene were used as representative SOA precursors in Beijing and Hyytiälä, respectively:

*Selection of SOA precursors*

*Monoterpenes have been found as major volatile organic compounds (VOC) in Hyytiälä, and α- and β-pinene are isomers accounting for > 60% of the total VOC (Kourtchev et al., 2008;Hakola et al., 2012). Previous studies found that both α- and β-pinene exhibit high SOA yield upon oxidation by $O_3$ (Lee et al., 2006;Zhang et al., 2015), and β-pinene SOA exhibits a higher potential in generating RS in water (Tong et al., 2016;Tong et al., 2017;Tong et al., 2018;Tong et al., 2019). To shorten the collection time of SOA and minimize the influence of aerosol aging on the aqueous RS detection, β-pinene was chosen as representative biogenic precursor for the SOA formation in Hyytiälä. In contrast, naphthalene has been found as key anthropogenic VOC in Beijing, which has high SOA yield upon photooxiation and potentially important contribution to SOA formation in Beijing (Chan et al., 2009;Huang et al., 2019). Recently studies also indicate that naphthalene SOA are redox-active substances and may play an important role in cytotoxicity of Beijing PM (Liu et al., 2020;Han et al., 2020). Thus, we choose naphthalene as representative anthropogenic precursor for the SOA formation in Beijing.*

For the sample treatment procedure of HA or FA water suspensions, why was the procedure necessary? So, does it mean that a water-soluble fraction of HA or FA was extracted and only analyzed?

Response: Yes, we only analyzed the water-soluble fraction of HA or FA. There are reasons for filtering out the insoluble fraction of HA or FA as discussed in the following. According to previous studies, humic-like substances frequently account for a small mass fraction of ambient $PM_{2.5}$ (< 10%, see Table S3). The solubility and chemical reactivity of these substances are largely unknown. The used HA or FA standard compounds can only be partly dissolved in water through a slow kinetic process. Thus, it is difficult to make an equivalent concentration of HA or FA suspension that can represent the water-soluble and insoluble fractions of humic-like substances contained in one ambient $PM_{2.5}$ sample. Moreover, only by keeping the initial concentration of aqueous-phase HA or FA known, the chemical reaction mechanism of the surrogate compounds can be understood quantitatively. Finally, the co-existence of solid phase and aqueous-phase HA or FA in the surrogate solutions will complicate the chemistry upon inducing surface adsorption and chemistry. To clarify the necessity to use water-soluble fraction of HA and FA in this study, we add the following text in lines 29-37 of SI:

*HA or FA are used in this study as standard surrogate compounds for humic-like substances and are known to only partly dissolve in water through a slow kinetic process (Baduel et al., 2009;Verma et al., 2015). Thus, it is difficult to generate an equivalent concentration of a HA or FA suspension that can represent the water-soluble and insoluble fractions of humic-like substances contained in one ambient $PM_{2.5}$ sample. Only by keeping the initial concentration of aqueous-phase HA or FA known, the chemical reaction mechanism of the surrogate compounds can be understood quantitatively. Beyond this, the co-existence of solid phase and aqueous-phase HA or FA in the surrogate solutions will complicate the chemistry of the surrogate compounds upon potentially inducing surface adsorption and surface chemistry effects. Thus, for simplicity, we only analyzed the water-soluble fraction of HA or FA in this study.*

Relative fractions of different radicals, particularly organic and OH radicals, seem to be dependent on the origin of PM samples. What would be the most important factor determining such relative fractions?

Response: Thank you for pointing out this important question. To clarify the PM source-dependent formation mechanisms of radicals, we add the following discussions in the main text (lines 282-293):

*We speculate that hydrolytic or thermal decomposition of ROOH may play a major role in the formation of RS by $PM_{2.5}$ from remote forest locations like Hyytiälä, where large fractions of peroxide-containing HOM have been detected (Mutzel et al., 2015;Tröstl et al., 2016;Tong et al., 2019;Pye et al., 2019;Roldin et al., 2019;Bianchi et al., 2019). ROOH can generate $^\bullet OH$ and O-centered organic radicals through decomposition (ROOH $\rightarrow RO^\bullet + {}^\bullet OH$) and Fenton-like reactions ($Fe^{2+} + ROOH \rightarrow Fe^{3+} + RO^\bullet + OH^-$; $Fe^{2+} + ROOH \rightarrow Fe^{3+} + RO^- + OH^\bullet$) (Tong et al., 2016). Interconversion of $RO^\bullet$, $R^\bullet$ and $ROO^\bullet$ radicals can lead to the formation of $O_2^{\bullet-}$ and $H_2O_2$ (Chevallier et al., 2004;Tong et al., 2018), which can further react with $Fe^{2+}$ to form $^\bullet OH$ ($Fe^{2+} + H_2O_2 \rightarrow Fe^{3+} + {}^\bullet OH + OH^-$). In $PM_{2.5}$ from urban areas, transition metal ions and HULIS are expected to play a major role in aqueous-phase formation and interconversion of $^\bullet OH$, $O_2^{\bullet-}$ and $H_2O_2$ (Lloyd et al., 1997;Valavanidis et al., 2000;Zheng et al., 2013;Hayyan et al., 2016;Lakey et al., 2016;Tan et al., 2016;Kuang et al., 2017;Ma et al., 2018;Li et al., 2019).*

In Figure 4, why were only transition metals shown in its absolute concentration, whereas the others were in the relative abundance (%)?

Response: The used ICP-MS method for quantification of water-soluble transition metals is well-established and accurate. Therefore, we can obtain reliable results on the abundances of target transition metals in Hyytiälä, Mainz, and Beijing PM as well as the absolute concentrations of metal ions in water extracts of these PM. However, it remains a challenge to use mass spectrometry in combination with HPLC techniques to differentiate and quantify aromatic or HOM mixtures in ambient $PM_{2.5}$, due to the various relationships of chromatographic area and concentrations of different aromatics or HOMs (Tong et al., 2019). Therefore, we cannot give the absolute concentration information of all aromatic compounds and HOMs. To clarify the different unit in Figure 4, we add the following text in lines 51-54 of SI:

*Due to the technique limitation, we are not able to quantify different aromatic or HOM species. In contrast, the method for quantification of water-soluble transition metals is well-established. Therefore, we can obtain reliable results on the absolute concentrations of target transition metals in Hyytiälä, Mainz, and Beijing PM or their water extracts.*

Lines 369-373: Why did naphthalene SOA have a lower radical yield than β-pinene SOA with the same mass concentration? This information should be important to generalize the findings of this study.

Response: To clarify the different mass-specific radical yields of naphthalene SOA and β-pinene SOA, we add the following text in lines 68-76 of SI:

*The mass-specific radical yield of laboratory-generated SOA is strongly dependent on the abundance of peroxide-containing highly oxygenated organic molecules (Tong et al., 2019), which are involved in radical formation upon thermal-, hydrolytic-, and photolytic- decompositions as well as Fenton-like reactions in water (Chen et al., 2011;Badali et al., 2015;Tong et al., 2016). Previous studies found that the mass fraction of organic peroxides in β-pinene SOA (42%) can be two times higher than in Naphthalene SOA (19-28%) (Kautzman et al., 2010;Tong et al., 2018). Our recent findings showed a positive correlation of HOM abundance and radical yields by both ambient PM and laboratory-generated SOA (Tong et al., 2019). Therefore, we suggest that the low abundance of peroxide-containing HOMs in naphthalene SOA is the major reason for its lower radical yield than β-pinene SOA.*

It seems that consistently the higher relative fraction of C-centered radicals is associated with the lower relative fraction of OH radicals. Does it suggest that OH radicals are consumed by something to generate C-centered radicals? This point deserves detailed discussions in the text.

Response: Thank you. We agree with that aqueous OH radicals may react with water-soluble organic compounds to form O-centered organic radicals, the rearrangement of which can form C-centered organic radicals (Chevallier et al., 2004). We noted this point by adding the following text in lines 270-271 of the main text:

*Moreover, rearrangement reactions in water can convert $RO^\bullet$ into $R^\bullet$ (Chevallier et al., 2004), which may warrant further investigation.*

When HA was replaced by FA, the dominant radical species becomes $O_2^-$ from C-centered radical. Do you any explanation on the different radical distribution?

Response: The major formation of $O_2^{\bullet-}$ in aqueous mixtures of CHP+$Fe^{2+}$+FA but C-centered radicals in CHP+$Fe^{2+}$+HA may reflect the different reactivities of FA from HA in Fenton-like reactions. To clarify this point, we add the following text in lines 408-412:

*Different reactivities of HA and FA are also reflected by the different RF values of $O_2^{\bullet-}$ and C-centered radicals observed at high concentrations of FA and HA (Figure 6h vs. Figure 6f) as well as in reactions mixtures with copper instead of iron ions (Figure S6). Further investigations will be required to resolve the underlying reaction mechanisms and kinetics.*

Lines 496-498: It is said that "the higher relative amount of detected radicals and $H_2O_2$ formed by urban $PM_{2.5}$ can see as a measure of higher potential oxidative damage", but the totally formed radical amount decreases with PM mass. The statement seems contradicting and needs clarification.

Response: Thank you. We replaced this sentence with the following text (lines 431-433):

*Overall, our findings show how the composition of $PM_{2.5}$ can influence the amount and nature of aqueous-phase RS, which may explain differences in the chemical reactivity and health effects of particulate matter in clean and polluted air.*

Minor points:

Line 236: The -> the

Response: We corrected this typo.

**References**

Badali, K., Zhou, S., Aljawhary, D., Antiñolo, M., Chen, W., Lok, A., Mungall, E., Wong, J., Zhao, R., and Abbatt, J.: Formation of hydroxyl radicals from photolysis of secondary organic aerosol material, Atmos. Chem. Phys., 15, 7831-7840, 2015.

Baduel, C., Voisin, D., and Jaffrezo, J.-L.: Comparison of analytical methods for Humic Like Substances (HULIS) measurements in atmospheric particles, Atmos. Chem. Phys., 9, 5949-5962, 2009.

Bianchi, F., Kurtén, T., Riva, M., Mohr, C., Rissanen, M. P., Roldin, P., Berndt, T., Crounse, J. D., Wennberg, P. O., and Mentel, T. F.: Highly oxygenated organic molecules (HOM) from gas-phase autoxidation involving peroxy radicals: A key contributor to atmospheric aerosol, Chem. Rev., 119, 3472-3509, 2019.

Chan, A. W. H., Kautzman, K., Chhabra, P., Surratt, J., Chan, M., Crounse, J., Kürten, A., Wennberg, P., Flagan, R., and Seinfeld, J.: Secondary organic aerosol formation from photooxidation of naphthalene and alkylnaphthalenes: implications for oxidation of intermediate volatility organic compounds (IVOCs), Atmos. Chem. Phys., 9, 3049-3060, 2009.

Chen, Q., Liu, Y., Donahue, N. M., Shilling, J. E., and Martin, S. T.: Particle-phase chemistry of secondary organic material: modeled compared to measured O: C and H: C elemental ratios provide constraints, Environ. Sci. Technol., 45, 4763-4770, 2011.

Chevallier, E., Jolibois, R. D., Meunier, N., Carlier, P., and Monod, A.: "Fenton-like" reactions of methylhydroperoxide and ethylhydroperoxide with $Fe^{2+}$ in liquid aerosols under tropospheric conditions, Atmos. Environ., 38, 921-933, 2004.

Hakola, H., Hellén, H., Hemmilä, M., Rinne, J., and Kulmala, M.: In situ measurements of volatile organic compounds in a boreal forest, Atmos. Chem. Phys., 12, 11665-11678, 2012.

Han, J., Wang, S., Yeung, K., Yang, D., Gu, W., Ma, Z., Sun, J., Wang, X., Chow, C.-W., and Chan, A. W.: Proteome-wide effects of naphthalene-derived secondary organic aerosol in BEAS-2B cells are caused by short-lived unsaturated carbonyls, Proc. Natl. Acad. Sci. U.S.A., 10.1073/pnas.2001378117, 2020.

Hayyan, M., Hashim, M. A., and AlNashef, I. M.: Superoxide ion: generation and chemical implications, Chem. Rev., 116, 3029-3085, 2016.

Huang, G., Liu, Y., Shao, M., Li, Y., Chen, Q., Zheng, Y., Wu, Z., Liu, Y., Wu, Y., Hu, M., Li, X., Lu, S., Wang, C., Liu, J., Zheng, M., and Zhu, T.: Potentially Important Contribution of Gas-Phase Oxidation of Naphthalene and Methylnaphthalene to Secondary Organic Aerosol during Haze Events in Beijing, Environ. Sci. Technol., 53, 1235-1244, 2019.

Kautzman, K., Surratt, J., Chan, M., Chan, A., Hersey, S., Chhabra, P., Dalleska, N., Wennberg, P., Flagan, R., and Seinfeld, J.: Chemical composition of gas-and aerosol-phase products from the photooxidation of naphthalene, J. Phys. Chem. A, 114, 913-934, 2010.

Kourtchev, I., Ruuskanen, T., Keronen, P., Sogacheva, L., Dal Maso, M., Reissell, A., Chi, X., Vermeylen, R., Kulmala, M., Maenhaut, W., and Claeys, M.: Determination of isoprene and α-/β-pinene oxidation products in boreal forest aerosols from Hyytiälä, Finland: diel variations and possible link with particle formation events, Plant Biol., 10, 138-149, 2008.

Kuang, X. M., Scott, J. A., da Rocha, G. O., Betha, R., Price, D. J., Russell, L. M., Cocker, D. R., and Paulson, S. E.: Hydroxyl radical formation and soluble trace metal content in particulate matter from renewable diesel and ultra low sulfur diesel in at-sea operations of a research vessel, Aerosol Sci. Technol., 51, 147-158, 2017.

Lakey, P. S., Berkemeier, T., Tong, H., Arangio, A. M., Lucas, K., Pöschl, U., and Shiraiwa, M.: Chemical exposure-response relationship between air pollutants and reactive oxygen species in the human respiratory tract, Sci. Rep., 6, 32916, 2016.

Lee, A., Goldstein, A. H., Keywood, M. D., Gao, S., Varutbangkul, V., Bahreini, R., Ng, N. L., Flagan, R. C., and Seinfeld, J. H.: Gas-phase products and secondary aerosol yields from the ozonolysis of ten different terpenes, J. Geophys. Res., 111, D07302, 2006.

Li, X., Han, J., Hopke, P. K., Hu, J., Shu, Q., Chang, Q., and Ying, Q.: Quantifying primary and secondary humic-like substances in urban aerosol based on emission source characterization and a source-oriented air quality model, Atmos. Chem. Phys., 19, 2327-2341, 2019.

Liu, F., Saavedra, M. G., Champion, J. A., Griendling, K. K., and Ng, N. L.: Prominent Contribution of Hydrogen Peroxide to Intracellular Reactive Oxygen Species Generated upon Exposure to Naphthalene Secondary Organic Aerosols, Environ. Sci. Technol. Lett., 7, 171-177, 2020.

Lloyd, R. V., Hanna, P. M., and Mason, R. P.: The origin of the hydroxyl radical oxygen in the Fenton reaction, Free Radical Biol. Med., 22, 885-888, 1997.

Ma, Y., Cheng, Y., Qiu, X., Cao, G., Fang, Y., Wang, J., Zhu, T., Yu, J., and Hu, D.: Sources and oxidative potential of water-soluble humic-like substances (HULIS WS) in fine particulate matter (PM$_{2.5}$) in Beijing, Atmos. Chem. Phys., 18, 5607-5617, 2018.

Mutzel, A., Poulain, L., Berndt, T., Iinuma, Y., Rodigast, M., Böge, O., Richters, S., Spindler, G., Sipilä, M., Jokinen, T., Markku, K., and Hartmut, H.: Highly oxidized multifunctional organic compounds observed in tropospheric particles: A field and laboratory study, Environ. Sci. Technol., 49, 7754-7761, 2015.

Pye, H. O., D'Ambro, E. L., Lee, B. H., Schobesberger, S., Takeuchi, M., Zhao, Y., Lopez-Hilfiker, F., Liu, J., Shilling, J. E., and Xing, J.: Anthropogenic enhancements to production of highly oxygenated molecules from autoxidation, Proc. Natl. Acad. Sci. U.S.A., 116, 6641-6646, 2019.

Roldin, P., Ehn, M., Kurtén, T., Olenius, T., Rissanen, M. P., Sarnela, N., Elm, J., Rantala, P., Hao, L., and Hyttinen, N.: The role of highly oxygenated organic molecules in the Boreal aerosol-cloud-climate system, Nat. Commun., 10, 1-15, 2019.

Tan, J., Xiang, P., Zhou, X., Duan, J., Ma, Y., He, K., Cheng, Y., Yu, J., and Querol, X.: Chemical characterization of humic-like substances (HULIS) in PM$_{2.5}$ in Lanzhou, China, Sci. Total Environ., 573, 1481-1490, 2016.

Tong, H., Arangio, A. M., Lakey, P. S., Berkemeier, T., Liu, F., Kampf, C. J., Brune, W. H., Pöschl, U., and Shiraiwa, M.: Hydroxyl radicals from secondary organic aerosol decomposition in water, Atmos. Chem. Phys., 16, 1761-1771, 2016.

Tong, H., Lakey, P. S., Arangio, A. M., Socorro, J., Kampf, C. J., Berkemeier, T., Brune, W. H., Pöschl, U., and Shiraiwa, M.: Reactive oxygen species formed in aqueous mixtures of secondary organic aerosols and mineral dust influencing cloud chemistry and public health in the Anthropocene, Faraday Discuss., 200, 251-270, 2017.

Tong, H., Lakey, P. S., Arangio, A. M., Socorro, J., Shen, F., Lucas, K., Brune, W. H., Pöschl, U., and Shiraiwa, M.: Reactive oxygen species formed by secondary organic aerosols in water and surrogate lung fluid, Environ. Sci. Technol., 52, 11642-11651, 2018.

Tong, H., Zhang, Y., Filippi, A., Wang, T., Li, C., Liu, F., Leppla, D., Kourtchev, I., Wang, K., Keskinen, H.-M., Levula, J. T., Arangio, A. M., Shen, F., Ditas, F., Martin, S. T., Artaxo, P., Godoi, R. H. M., Yamamoto, C. I., Souza, R. A. F. d., Huang, R.-J., Berkemeier, T., Wang, Y., Su, H., Cheng, Y., Pope, F. D., Fu, P., Yao, M., Pöhlker, C., Petäjä, T., Kulmala, M., Andreae, M. O., Shiraiwa, M., Pöschl, U., Hoffmann, T., and Kalberer, M.: Radical Formation by Fine Particulate Matter Associated with Highly Oxygenated Molecules, Environ. Sci. Technol., 53, 12506-12518, 2019.

Tröstl, J., Chuang, W. K., Gordon, H., Heinritzi, M., Yan, C., Molteni, U., Ahlm, L., Frege, C., Bianchi, F., and Wagner, R.: The role of low-volatility organic compounds in initial particle growth in the atmosphere, Nature, 533, 527-531, 2016.

Valavanidis, A., Salika, A., and Theodoropoulou, A.: Generation of hydroxyl radicals by urban suspended particulate air matter. The role of iron ions, Atmos. Environ., 34, 2379-2386, 2000.

Verma, V., Wang, Y., El-Afifi, R., Fang, T., Rowland, J., Russell, A. G., and Weber, R. J.: Fractionating ambient humic-like substances (HULIS) for their reactive oxygen species activity–Assessing the importance of quinones and atmospheric aging, Atmos. Environ., 120, 351-359, 2015.

Zhang, X., McVay, R. C., Huang, D. D., Dalleska, N. F., Aumont, B., Flagan, R. C., and Seinfeld, J. H.: Formation and evolution of molecular products in α-pinene secondary organic aerosol, Proc. Natl. Acad. Sci. U.S.A., 112, 14168-14173, 2015.

Zheng, G., He, K., Duan, F., Cheng, Y., and Ma, Y.: Measurement of humic-like substances in aerosols: A review, Environ. Pollut., 181, 301-314, 2013.